# Adaptive Exploration for Multi-Reward Multi-Policy Evaluation

**Alessio Russo** [1]  **Aldo Pacchiano** [1 2]

## Abstract

We study the policy evaluation problem in an online multi-reward multi-policy discounted setting, where multiple reward functions must be evaluated simultaneously for different policies. We adopt an $(\epsilon, \delta)$-PAC perspective to achieve $\epsilon$-accurate estimates with high confidence over finite or convex sets of rewards, a setting that has not been systematically studied in the literature. Building on prior work on Multi-Reward Best Policy Identification, we adapt the `MR-NaS` exploration scheme (Russo & Vannella, 2024) to jointly minimize sample complexity for evaluating different policies across different reward sets. Our approach leverages an instance-specific lower bound revealing how the sample complexity scales with a measure of value deviation, guiding the design of an efficient exploration policy. Although computing this bound entails a hard non-convex optimization, we propose an efficient convex approximation that holds for both finite and convex reward sets. Experiments in tabular domains demonstrate the effectiveness of this adaptive exploration scheme. Code repository: https://github.com/rssalessio/mu lti-reward-multi-policy-eval.

## 1. Introduction

This paper investigates methods for efficiently evaluating one or more policies across various reward functions in an online discounted Markov Decision Process (MDP) (Puterman, 2014), a key challenge in Reinforcement Learning (RL) (Sutton & Barto, 2018), where the aim is to compute the value of each policy. Accurate value estimation serves many purposes, from verifying a policy's effectiveness to providing insights for policy improvement.

[1]Faculty of Computing and Data Sciences, Boston University, Boston (MA), USA. [2]Broad Institute of MIT and Harvard, Boston (MA), USA. Correspondence to: Alessio Russo <arusso2@bu.edu>.

*Proceedings of the 42$^{nd}$ International Conference on Machine Learning*, Vancouver, Canada. PMLR 267, 2025. Copyright 2025 by the author(s).

There are several applications where one has multiple policies to evaluate, e.g., multiple policies arising from using different hyperparameters (Dann et al., 2023; Chen et al., 2024; Poddar et al., 2024). Similarly, multiple reward functions often arise in real-world decision-making problems, making it crucial to evaluate how a policy performs across diverse objectives. As an example, large language models (Brown et al., 2020) are fine-tuned on human feedback that spans a wide range of user preferences and goals (Ziegler et al., 2019; Rafailov et al., 2024; Poddar et al., 2024), effectively producing multiple distinct reward signals. Other applications, similarly, involve multiple rewards, such as: user-preference modeling, robotics tasks aiming to reach different goals, or intent-based radio network optimization (Nahum et al., 2023; De Alwis et al., 2023; Russo & Vannella, 2024; Poddar et al., 2024).

In general, it is challenging to efficiently and accurately evaluate a policy over multiple objectives, potentially for multiple policies aimed at solving different tasks (Sutton et al., 2011; McLeod et al., 2021; Jain et al., 2024). Indeed, when multiple policies and distinct reward sets are involved, it is not obvious how best to gather data in a way that balances efficiency and accuracy.

Prior research approached this issue in different ways. One direction aims to minimize the mean squared error (MSE) of the value estimator (or the variance of the importance sampling estimator) to guide adaptive exploration. In (Hanna et al., 2017), this is done for single-reward policy evaluation. In (Mukherjee et al., 2022) they study single-reward policy evaluation in tree MDPs, and propose a variance-driven behavior policy that minimizes the MSE. Similarly, Jain et al. (2024) propose a variance-driven exploration scheme for a finite collection of policy–reward pairs. However, these methods may not always guarantee sample-efficient exploration or provide $(\epsilon, \delta)$-PAC guarantees on the sample complexity. Weissmann et al. (2025) recently introduced a sample-efficient behavior-policy design for evaluating multiple target policies and proved $(\epsilon, \delta)$ excess-risk guarantees; their results, however, apply only to single-reward stochastic bandit models (Lattimore & Szepesvári, 2020). In (Dann et al., 2023; Chen et al., 2024) the authors address multi-policy evaluation for a single reward under the $(\epsilon, \delta)$-PAC criteria in episodic MDPs. However, their sample complexity guarantees are instance-dependent in the transition

dynamics (e.g., dependent on state-action visitation structure) but not in the reward. Consequently, these bounds do not characterize how the interaction between rewards and transitions—such as sparse rewards under specific dynamics—affects evaluation complexity.

To investigate the problem of devising an exploration strategy for this setting, we study the sample complexity of multi-reward multi-policy evaluation, and adopt an $(\epsilon, \delta)$-PAC viewpoint to achieve $\epsilon$-accurate estimates with confidence $1 - \delta$ over finite or convex sets of rewards. Furthermore, while prior works focus on estimating the expected value of a policy under the initial state distribution, our work evaluates policies across all states (a value vector). This broader scope is critical for applications requiring reliable verification of policy behavior and enabling explainable RL (Puiutta & Veith, 2020; Ruggeri et al., 2025).

Our approach optimizes the behavior policy to maximize the evidence gathered from the environment at each time-step, and builds on techniques from the Best Policy Identification (BPI) literature (Al Marjani et al., 2021), which itself draws inspiration from Best Arm Identification techniques in bandit problems (Garivier & Kaufmann, 2016; Kaufmann et al., 2016; Russo & Proutiere, 2023b; Russo et al., 2025). These methods cast the problem of finding the optimal policy as a hypothesis testing problem. The core insight is to establish an instance-specific sample complexity lower bound—formulated as an optimization problem—where its solution directly yields the optimal exploration distribution.

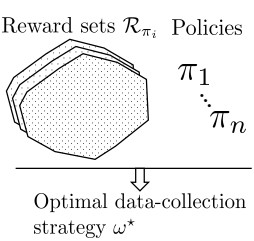

Reward sets $\mathcal{R}_{\pi_i}$ Policies

$\pi_1$
$\cdots$
$\pi_n$

Optimal data-collection strategy $\omega^\star$

Building on insights from Multi-Reward BPI (Russo & Vannella, 2024), which extends the BPI framework to multiple rewards, we develop an instance-specific sample complexity lower bound for multi-reward multi-policy evaluation, dependent on the dynamics and the rewards. To our knowledge, this constitutes the first such bound even for the single-policy single-reward case. Furthermore, for convex reward sets, we present an alternative sample complexity characterization compared to (Russo & Vannella, 2024), which yields a closed-form solution when the set of rewards includes all possible rewards.

Our result exhibits a sample complexity that scales with the worst-case policy-reward pair—rather than the sum over individual policy-reward pairs—capturing the inherent difficulty of efficiently evaluating all combinations simultaneously. Particularly, the complexity scales according to a measure of value deviation $\rho_r^\pi(s, s') := V_r^\pi(s') -$

$\mathbb{E}_{\hat{s} \sim P(s, \pi(s))}[V_r^\pi(\hat{s})]$, similar to the variance of the value:

$$O\left( \sup_{\pi \in \Pi, r \in \mathcal{R}_\pi} \max_{s, s' \in \mathcal{S}} \frac{\gamma^2 |\rho_r^\pi(s, s')|^2}{\epsilon^2 (1 - \gamma)^2 \omega^\star(s, \pi(s))} \right),$$

where $\omega^\star$ is the stationary distribution induced by the exploration policy [1].

Finally, we adapt MR-NaS (Russo & Vannella, 2024), an extension of NaS (Al Marjani et al., 2021), to our setting, and prove its asymptotic optimality for policy evaluation up to a constant factor. We further illustrate its practical efficiency through experiments in various tabular environments.

## 2. Related Work

Reinforcement Learning (RL) exploration techniques have typically focused only on the problem of learning the optimal policy for a single objective (Sutton & Barto, 2018). This domain has generated a vast body of work, often inspired by the multi-armed bandit literature (Lattimore & Szepesvári, 2020), with approaches ranging from $\epsilon$-greedy and Boltzmann exploration (Watkins, 1989; Sutton & Barto, 2018) to more sophisticated methods based on Upper-Confidence Bounds (UCB) (Auer, 2002), Bayesian procedures (Osband et al., 2013; Russo et al., 2018) or Best Policy Identification techniques (Al Marjani et al., 2021; Wagenmaker et al., 2022; Taupin et al., 2023).

Despite these advances, the challenge of designing exploration strategies for online policy evaluation has received comparatively little attention. Early work in this direction examined multi-armed bandit problems (Antos et al., 2008) and function evaluation (Carpentier & Munos, 2012), showing that efficient exploration requires allocating more samples where variance is higher. Linke et al. (2020) considered a bandit setting with a finite number of tasks, and focused on minimizing the mean squared error. Still within the bandit setting, Weissmann et al. (2025) recently proposed a behavior-policy scheme for evaluating $n$ target policies and established $(\epsilon, \delta)$ excess-risk guarantees. For $\sigma^2$-sub-Gaussian rewards, their sample-complexity bound scales as $O\left( \frac{\sigma^2 w^\star \log(n/\delta)}{\epsilon^2} \right)$, where $w^\star$ is the maximal importance-sampling weight (upper-bounded by $n$).

For MDPs, Hanna et al. (2017) introduced the idea of optimizing the behavior policy (i.e., the exploration strategy) by directly minimizing the variance of importance sampling—one of the earliest efforts to design an exploration strategy specifically for policy evaluation in MDPs. In (Mukherjee et al., 2022) the authors studied single-reward policy evaluation in tree MDPs, and proposed a variance-driven behavior policy that minimizes the MSE. Separately, (McLeod et al., 2021) proposed SF-NR, which uses the Suc-

---

[1]In this work, we use the terms "exploration policy" and "behavior policy" interchangeably.

cessor Representation framework (Dayan, 1993) to guide exploration for value estimation over a finite set of tasks. Additionally, in another line of research, Papini et al. (2024) analyzed the behavioral policy optimization problem for policy gradient methods. More closely related to our work is (Jain et al., 2024), where the authors tackled the evaluation of a finite set of policy–reward pairs and proposed `GVFExplorer`, an adaptive exploration method that optimizes the behavior policy with the goal of minimizing the estimation MSE. Their resulting exploration policy is variance-driven, and resembles that derived by Mukherjee et al. (2022). However, these works do not provide PAC guarantees, nor directly provide a strategy that directly minimizes the sample complexity.

In (Dann et al., 2023) they investigated the multi-policy single-reward evaluation problem in a PAC framework. They proposed an on-policy method that leverages the fact that the policies may overlap in a significant way. In (Chen et al., 2024) the authors designed a behavior policy that optimizes the coverage of the target policy set. However, these works do not provide instance-dependent bounds in terms of the value of the target policies, nor study the multi-reward policy evaluation problem.

## 3. Problem Setting

In this section, we describe the MDP model considered and the policy evaluation setting.

### 3.1. Markov Decision Processes (MDPs)

Markov Decision Processes (MDPs) are widely utilized to model sequential decision-making tasks (Puterman, 2014). In these tasks, at each time-step $t = 1, 2, \ldots$ an agent observes the current state of the MDP $s_t$ and selects an action $a_t$ to achieve a desired objective. This objective is encapsulated in terms of a reward $r_t \in [0, 1]$, observed after selecting the action. In RL, the primary goal of the agent is to determine a sequence of actions that maximizes the total reward collected from an unknown MDP.

**Discounted MDPs.** We consider a discounted MDP (Puterman, 2014) of the type $M = (\mathcal{S}, \mathcal{A}, P, \gamma)$, where: $\mathcal{S}$ is a finite state space of size $S = |\mathcal{S}|$; $\mathcal{A}$ is a finite action space of size $A = |\mathcal{A}|$; $P : \mathcal{S} \times \mathcal{A} \to \Delta(\mathcal{S})$ is the transition function, which maps state-action pairs to distributions over states; lastly, $\gamma \in (0, 1)$ is the discount factor. In the following we consider also reward functions $r : \mathcal{S} \times \mathcal{A} \to [0, 1]$ that are bounded and deterministic functions of state-action pairs, and write $M_r = (M, r)$ to denote the MDP $M$ with reward $r$. In classical RL, an agent is interested in maximizing the total discounted reward collected from $M_r$: $r_1 + \gamma r_2 + \gamma^2 r_3 + \ldots$. The problem of computing an optimal sequence of actions (i.e., that maximizes the total discounted reward) can be reduced to that of finding

a Markovian policy $\pi : \mathcal{S} \to \Delta(\mathcal{A})$ that maps states to distributions over actions (Puterman, 2014). For a Markovian policy $\pi$ we define the discounted value of $\pi$ in $M_r$ at state $s$ as $V_r^\pi(s) := \mathbb{E}^\pi[\sum_{t \geq 1} \gamma^{t-1} r(s_t, a_t) | s_1 = s]$, where $s_{t+1} \sim P(\cdot | s_t, a_t)$ and $a_t \sim \pi(\cdot | s_t)$ (if $\pi$ is deterministic, we write $a_t = \pi(s_t)$). We also write $V_r^\pi$ and omit the dependency on $M$ when it is clear from the context. We also define $V_r^\star(s) := \max_\pi V_r^\pi(s)$ to be the optimal value in $s$ over all Markovian policies. We further define the action-value function in $M_r$ as $Q_r^\pi(s, a) := r(s, a) + \gamma \mathbb{E}_{s' \sim P(\cdot | s, a)}[V_r^\pi(s')]$.

**Additional notation.** For a set $\mathcal{U} \subset \mathbb{R}^n, n \in \mathbb{N}$, we denote by $\overline{\mathcal{U}}$ its closure, and we also define $\Delta(\mathcal{U})$ to be the set of distributions over $\mathcal{U}$. We also denote by $N_t(s, a) = \sum_{n=1}^t \mathbf{1}_{((s_n, a_n) = (s, a))}$ the number of times an algorithm has visited a state-action pair $(s, a)$ up to time $t$ (sim. we define $N_t(s) = \sum_a N_t(s, a)$). For two MDPs $M, M'$ with transition functions, respectively, $P, P'$, we write $\mathrm{KL}_{P|P'}(s, a)$ to indicate the KL-divergence between the two transition functions in $(s, a) \in \mathcal{S} \times \mathcal{A}$ (we also write $\mathrm{KL}_{M|M'}(s, a)$). We denote by $\mathrm{kl}(x, y) = x \log(x/y) + (1 - x) \log((1 - x)/(1 - y))$ the Bernoulli KL-divergence between two parameters $x, y \in [0, 1]$.

**Assumptions.** We consider the problem of evaluating a *finite set of deterministic target policies* $\Pi = \{\pi_1, \ldots, \pi_N\}$ over finite, or possibly convex, reward sets (we discuss more on this in the next paragraph). On the MDP $M$, we impose the following assumption.

**Assumption 3.1.** *The MDP $M$ is communicating and aperiodic under a uniform policy, and the learner has no prior knowledge of $P$. The starting state is also arbitrary.*

As in (Al Marjani et al., 2021; Russo & Vannella, 2024), we assume that the MDP is communicating, which avoids the awkward case where the algorithm could enter a subclass of states from which there is no possible comeback, and thus it becomes impossible to identify the value of a policy $\pi$ to the desired accuracy. Lastly, the assumption of aperiodicity can be met by assuming that the exploration policy assigns positive probability to some state-action pair $(s, a)$ satisfying $P(s|s, a) > 0$. Such assumption is needed to ensure ergodicity of the chain induced by the exploration policy.

**Set of Rewards.** In the following, we consider sets of reward functions $\mathcal{R}$ that are either finite, or convex, and assume that these sets are known to the agent beforehand. For tabular MDPs (i.e., with finite state-action spaces), assuming a natural labeling of the states $\mathcal{S} = \{s_1, \ldots, s_S\}$, we represent a reward $r$, for a policy $\pi$, as a vector in $[0, 1]^S$ (for simplicity, we omit the dependency on $\pi$). The $i$-th element of $r$ corresponds to the reward $r(s_i, \pi(s_i))$. For $\pi$, we indicate the canonical reward set by $\mathcal{R}_{\mathrm{canon}}^\pi = \{e_1, \ldots, e_S\}$ in $\mathbb{R}^S$, with $e_i$ being the $i$-th vector of the canonical basis, defined as $(e_i)_j = \mathbf{1}_{(i=j)}, \; i, j \in \{1, \ldots, S\}$. Lastly, we

denote the $\mathcal{R} = [0,1]^S$ case as reward-free since it encompasses all possible rewards for $\pi$.

### 3.2. Online Multi-Reward Multi-Policy Evaluation

In an online single-reward Policy Evaluation (PE) setting, the objective is to learn the value vector of a single policy–reward pair $(\pi, r)$. In the multi-reward, multi-policy case, we aim instead to learn the value vectors for $\Theta := \{(\pi, \mathcal{R}_\pi) : \pi \in \Pi\}$, meaning each policy $\pi \in \Pi$ is evaluated on every reward in its own reward set $\mathcal{R}_\pi$, which can differ across policies.

This setting is closely related to off-policy policy evaluation (OPE) (Thomas & Brunskill, 2016; Precup, 2000). However, OPE does not typically focus on the problem of optimizing the data collection policy. In this work, we study the problem of devising an optimal data-collection policy that, by online interactions with the MDP $M$, permits the agent to learn the value of pairs $(\pi, \mathcal{R}_\pi) \in \Theta$ as quickly as possible up to the desired accuracy.

**Online Multi-Reward Multi-Policy Evaluation.** We formalize our objective using the $(\epsilon, \delta)$-PAC (Probably Approximately Correct) framework. In such framework, an algorithm Alg interacts with $M$ until sufficient data has been gathered to output the value of $\pi$ up to $\epsilon$ accuracy for any reward $r \in \mathcal{R}_\pi$, for all $\pi \in \Pi$, with confidence $1 - \delta$.

Formally, an online PE algorithm Alg consists of:

- a *sampling rule* $(a_t)_{t \geq 1}$: upon observing a state $s_t$, Alg selects an action $a_t$, and then observes the next state $s_{t+1} \sim P(\cdot|s_t, a_t)$.

- a *stopping rule* $\tau$ that dictates when to stop the data collection process. $\tau$ is a stopping rule w.r.t. the filtration $(\mathcal{F}_t)_{t \geq 1}$, where $\mathcal{F}_t = \sigma(\{s_1, a_1, \ldots, a_{t-1}, s_t\})$ is the $\sigma$-algebra generated by the random observations made under Alg up to time $t$.

- an *estimated value* $\hat{V}_r^\pi$: at the stopping time $\tau$, Alg returns the estimated value $\hat{V}_r^\pi$ of the policy $\pi \in \Pi$ in $M$ for any chosen reward $r \in \mathcal{R}_\pi$.

Denoting by $\mathbb{P}_M$ (resp. $\mathbb{E}_M$) the probability law (resp. expectation) of the data observed under Alg in $M$, we define an algorithm to be $(\epsilon, \delta)$-PAC as follows.

**Definition 3.2** ($(\epsilon, \delta)$-PAC algorithm). An algorithm Alg is said to be multi-reward multi-policy $(\epsilon, \delta)$-PAC if, for any MDP $M$, policies-rewards set $\Theta$, $\delta \in (0, 1/2)$ and $\epsilon \in (0, \frac{1}{2(1-\gamma)})$, we have $\mathbb{P}_M[\tau < \infty] = 1$ (the algorithm stops almost surely) and

$$\mathbb{P}_M\left[\exists \pi \in \Pi, \exists r \in \mathcal{R}_\pi : \|V_r^\pi - \hat{V}_r^\pi\|_\infty > \epsilon\right] \leq \delta. \quad (1)$$

In other words, with probability at least $1 - \delta$, the algorithm's estimate $\hat{V}_r^\pi$ is within $\epsilon$ of $V_r^\pi$ for every $r \in \mathcal{R}_\pi, \pi \in \Pi$.

In the following section, we investigate the sample complexity of this problem and determine the minimal number of samples required to achieve the $(\epsilon, \delta)$-PAC guarantees. Our analysis reveals that the problem can be more challenging under certain rewards. In fact, the sample complexity is governed by the most difficult policy-reward pair in $\Theta$, rather than the sum of the individual complexities across pairs.

## 4. Adaptive Exploration through Minimal Sample Complexity

We seek to design a data collection strategy achieving minimal sample complexity. Building on BPI techniques (Garivier & Kaufmann, 2016; Al Marjani et al., 2021), we first derive an instance-specific sample complexity lower bound for any $(\epsilon, \delta)$-PAC algorithm.

This bound, which is posed as an optimization problem, specifies the optimal exploration policy, enabling the derivation of an efficient algorithm. The key step lies in bounding the expected log-likelihood ratio between the true model $M$ and a carefully constructed "*confusing*" model $M'$.

### 4.1. Set of Alternative Models

Confusing models are alternative models that are "similar" to $M$, but differ in certain key properties. As we see later, the set of alternative models is crucial for establishing a lower bound on the sample complexity $\tau$. To prove the lower bound, we frame the sample complexity problem as a goodness-of-fit test: does the observed data better align with the true model $M$ or an alternative model $M'$?

We seek a model $M'$ that is *confusing*—that is, it is "statistically" close to $M$ (in the KL sense), yet differs by at least $2\epsilon$ in the value of a policy $\pi$ under some reward $r$ (we discuss later why we choose $2\epsilon$ and not $\epsilon$). To find $M'$, we first construct the set of alternative models. Formally, for a policy $\pi \in \Pi$, reward $r \in \mathcal{R}_\pi$, the set of alternative models in $(\pi, r)$ is defined as

$$\text{Alt}_{\pi, r}^\epsilon(M) := \{M_r' : M \ll M_r', \ \|V_{M_r}^\pi - V_{M_r'}^\pi\|_\infty > 2\epsilon\},$$

where $M_r' = (\mathcal{S}, \mathcal{A}, P_r', r, \gamma)$ is an alternative MDP, and we index the transition function $P_r'$ by the corresponding reward $r$ for clarity. The notation $M \ll M_r'$ means that $P(s, a)$ is absolutely continuous with respect to $P_r'(s, a)$ for all $(s, a)$ (and $P$ is the transition function of $M$). We also denote by $\text{Alt}^\epsilon(M) = \cup_{\pi \in \Pi, r \in \mathcal{R}_\pi} \text{Alt}_{\pi, r}^\epsilon(M)$ the entire set of alternative models over $\mathcal{R}$.

From a sample complexity perspective, we can interpret this set as follows: a larger set of alternative models may lead to increased sample complexity since it becomes more likely that one of these models is statistically close to $M$ as measured by their KL divergence. Hence, we should expect the learning complexity to be dominated by the "worst" among such models.

Note that we require $\|V_{M_r}^\pi - V_{M_r'}^\pi\|_\infty > 2\epsilon$, rather than just $\epsilon$. If the separation were only $\epsilon$, an estimator $\hat{V}_r^\pi$ satisfying $\|\hat{V}_r^\pi - V_{M_r}^\pi\|_\infty \le \epsilon$ could also be simultaneously $\epsilon$-accurate for an alternative model $M'$ satisfying $\|V_{M_r}^\pi - V_{M_r'}^\pi\|_\infty > \epsilon$, making the models indistinguishable. Requiring $2\epsilon$ separation prevents this ambiguity. While this choice potentially weakens the resulting lower bound, we believe the looseness is at most a constant factor (see Theorem 5.2).

Lastly, consider the case where the set $\mathrm{Alt}_{\pi,r}^\epsilon(M)$ is empty for some pair $(\pi, r)$. In this situation, any model $P'$ sharing the same support as $P$ yields a value that is $2\epsilon$-close to the true value. Thus, the learning challenge for that reward is rather minimal. Therefore, characterizing when these sets are empty is crucial in the analysis. In the following, we denote by $\mathcal{R}_\pi^\epsilon = \{r \in \mathcal{R}_\pi : \mathrm{Alt}_{\pi,r}^\epsilon(M) \neq \emptyset\}$ the set of rewards for which the corresponding set of confusing models is non-empty.

**Value deviation.** To analyze these confusing sets, and their implications for sample complexity, we define the following instance-dependent quantity, that we refer to as the *one-step value deviation*:

$$\rho_r^\pi(s, s') := V_r^\pi(s') - \mathbb{E}_{\hat{s} \sim P(s, \pi(s))}[V_r^\pi(\hat{s})] \quad \forall s, s' \in \mathcal{S}.$$

This quantity measures how much the value at a state $s'$ deviates from the expected value under $\pi$ at $s$. As we see later, it is "easier" to construct alternative models if $|\rho_r^\pi(s, s')|$ is large. We also define these quantities in vector form $\rho_r^\pi(s) := \begin{bmatrix} \rho_r^\pi(s, s_1) & \dots & \rho_r^\pi(s, s_S) \end{bmatrix}^\top$, so that $\|\rho_r^\pi(s)\|_\infty = \max_{s'} |\rho_r^\pi(s, s')|$ is the maximum one-step deviation at $s$. The deviation $\rho_r^\pi$ is closely related to the *span* of the value function $\mathrm{sp}(V_r^\pi) := \max_{s'} V_r^\pi(s') - \min_s V_r^\pi(s)$, but, is in general smaller than the span (see Lemma B.3).

Using this measure of value deviation $\rho_r^\pi$, we are able to provide necessary and sufficient conditions under which $\mathrm{Alt}_{\pi,r}^\epsilon(M)$ is empty or not (see proof in Appendix B.1.1).

---

**Proposition 4.1.** *Fix a policy $\pi \in \Pi$ and a reward $r \in \mathcal{R}_\pi$. We have the following conditions: (1; necessary) if $\mathrm{Alt}_{\pi,r}^\epsilon(M) \neq \emptyset$ then $\exists s \in \mathcal{S} : \|\rho_r^\pi(s)\|_\infty > \frac{\epsilon(1-\gamma)}{\gamma}$; (2; sufficient) if $\exists s \in \mathcal{S} : \|\rho_r^\pi(s)\|_\infty > \frac{2\epsilon}{\gamma}$, then $\mathrm{Alt}_{\pi,r}^\epsilon(M) \neq \emptyset$.*

---

The proposition offers key insights into the challenges involved in learning the value function:

- As $\epsilon$ increases or $\gamma$ decreases, the necessary condition in Proposition 4.1 is less likely to be satisfied, potentially leading to a decrease in the number of alternative models (which suggests a smaller sample complexity).
- The proof highlights the concept of *confusing states*: a state $s$ is considered confusing if $\|\rho_r^\pi(s)\|_\infty >$

$\frac{2\epsilon}{\gamma}$. States with smaller values of $\|\rho_r^\pi(s)\|_\infty$ do not strongly affect the sample complexity. Additionally, Lemma B.3 (in the appendix) indicates that the maximum $\max_{s'} |\rho_r^\pi(s, s')|$ is typically attained at $s' \neq s$.

- Finally, if $\max_{s \in \mathcal{S}} \|\rho_r^\pi(s)\|_\infty = 0$, the set $\mathrm{Alt}_{\pi,r}^\epsilon(M)$ is empty for any values of $\epsilon$ and $\gamma$, implying no confusing models exist under such conditions.

Regarding the last point, in the following proposition, proved in Appendix B.1.3, we provide a sufficient and necessary condition for $\max_s \|\rho_r^\pi(s)\|_\infty = 0$.

---

**Proposition 4.2.** *The vectors $r$ for which $\max_s \|\rho_r^\pi(s)\|_\infty = 0$ is precisely the set $\{\alpha \mathbf{1} : \alpha \in [0, 1]\}$, where $\mathbf{1}$ is the vector of ones.*

---

While it may seem obvious that the all-ones reward cannot produce an alternative model, it is noteworthy that no other reward satisfies $\max_s \|\rho_r^\pi(s)\|_\infty = 0$.

We are now ready to discuss the sample complexity, and we refer the reader to Appendix B.1.3 for further discussion on the set of alternative models, including a characterization of when $\rho_r^\pi(s, s)$ is identically zero across all states.

## 4.2. Sample Complexity Lower Bound

As a consequence of Proposition 4.1, the analysis of the sample complexity must necessarily take into account the reward sets. However, for clarity, we adopt the assumption that for every state there exists a confusing model (see Prop. 4.1). While not strictly necessary (one could work with $\mathcal{R}_\pi^\epsilon$ instead), this assumption simplifies our analysis.

**Assumption 4.3.** For every state $s \in \mathcal{S}$ there exists $\pi \in \Pi, r \in \mathcal{R}_\pi$ such that $\|\rho_r^\pi(s)\|_\infty > \frac{2\epsilon}{\gamma}$.

To derive the sample complexity lower bound, we define the *characteristic time* $T_\epsilon(\omega; M)$ of a stationary state-action distribution $\omega$ under $M$:

$$T_\epsilon(\omega; M)^{-1} := \inf_{\pi \in \Pi, r \in \mathcal{R}_\pi, M_r' \in \mathrm{Alt}_{\pi,r}^\epsilon(M)} \mathbb{E}_\omega[\mathrm{KL}_{P|P_r'}(s, a)], \tag{2}$$

where $(s, a) \sim \omega$. In this optimization problem, we seek an alternative model $M'$ that is *confusing*, i.e., that minimizes the statistical difference from the true process $M$ under some pair $(\pi, r)$. Conversely, the sampling distribution $\omega$ serves to gather evidence distinguishing $M$ from any such alternative. Hence, the reciprocal $T_\epsilon(\omega; M)^{-1}$ can be interpreted as the *information-gathering rate* under $\omega$ in $M$.

Consequently, an optimal exploration strategy chooses $\omega$ to maximize this rate. From this perspective, one can show that, asymptotically, the sample-complexity lower bound for any $(\epsilon, \delta)$-PAC algorithm scales as

$$T_\epsilon^\star(M) = \inf_{\omega \in \Omega(M)} T_\epsilon(\omega; M), \tag{3}$$

where we denote by $\omega_{\text{opt}} \in \overline{\Omega}(M)$ a solution to the optimization in Eq. (3), and $\Omega(M)$ is defined as the following set of stationary state-action distributions: $\Omega(M) := \left\{ \omega \in \Delta(\mathcal{S} \times \mathcal{A}) : (\sum_a \omega(s,a) = \sum_{s',a'} P(s|s',a')\omega(s',a') \forall s)$ and $\omega(s,a) > 0 \ \forall (s,a) \right\}$. We have the following asymptotic lower bound, and the proof of the theorem can be found in Appendix B.2.1.

**Theorem 4.4.** *Under Assumption 4.3, for any $(\epsilon, \delta)$-PAC algorithm we have*

$$\liminf_{\delta \to 0} \frac{\mathbb{E}[\tau]}{\log(1/\delta)} \geq T^\star_\epsilon(M). \tag{4}$$

As discussed earlier, an exploration strategy matching the information gain $T^\star_\epsilon(M)$ maximizes the evidence collected per time-step required to distinguish between the true model $M$ and a confusing one.

This problem can be framed conceptually as a zero-sum game: the explorer seeks to maximize $T_\epsilon(\omega; M)^{-1}$ over $\omega$, while an adversary selects a confusing model for some policy–reward pair to minimize this value. Consequently, the sample complexity is determined by the most difficult policy–reward pair to discriminate from the true model.

However, it may seem counterintuitive to use an exploration strategy that depends on the true model itself $M$, since it is unknown. In practice, we use the empirical estimate of the MDP $M_t$ at time-step $t$ to derive an exploration strategy. We defer this discussion to Section 5.

**Optimality of behavior policies.** A natural question is whether we can use this result to identify when an exploration policy is suboptimal. For instance, consider the MDP in Figure 1 and the single target policy $\pi(\cdot|s) = a_2 \ \forall s$. For that MDP, it is *in general* sub-optimal to sample according to such policy, since under $\pi$ state $s_1$ becomes transient. In the following lemma we prove that a necessary condition for an optimal exploration policy is to guarantee that states with large deviation gap are visited infinitely often.

**Lemma 4.5.** *Let $T^{\pi_e}$ be the set of transient states (i.e., not recurrent) under a behavior policy $\pi_e$. Assume that $T^{\pi_e} \neq \emptyset$. If there exists $s_c \in T^{\pi_e}, \pi \in \Pi, r \in \mathcal{R}_\pi$ such that $\|\rho^\pi_r(s_c)\|_\infty > 2\epsilon/\gamma$, then $\pi_e$ is not optimal, in the sense that*

$$T_\epsilon(d^{\pi_e}; M)^{-1} = 0, \tag{5}$$

*for any stationary distribution $d^{\pi_e}$ (of the chain $P^{\pi_e}$) induced by $\pi_e$.*

The proof for this lemma can be found in Appendix B.2.2.

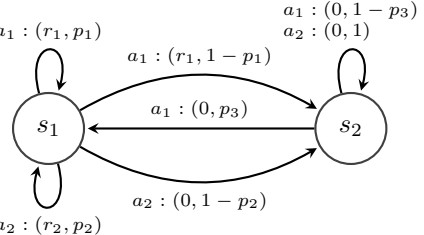

Figure 1. In this MDP (v. Example 4.6), in each edge it is indicated the action and the corresponding reward and transition probability.

This result is general, and also applies to single-policy single-reward cases. Moreover, we understand that an optimal exploration policy needs to frequently sample states with large deviation gaps $\|\rho^\pi_r(s)\|_\infty$. This intuition will be used to devise an exploration policy.

**Scaling.** Last, but not least, the lower bound expression does not clearly reveal its scaling behavior or its connection to the value deviation $\rho$. While we have briefly touched upon this relationship, these aspects are explored in greater depth in the next section, where a convexification of the problem provides further insights into these properties.

### 4.2.1. CONVEXITY OF THE LOWER BOUND

We find that it is hard to directly optimize $T_\epsilon(\omega; M)$, since the optimization over the set of alternative models may be non-convex. Observe that the set $\text{Alt}^\epsilon_{\pi,r}(M)$ can be seen as the union of two sets $\{M' : \max_s V^\pi_{M_r}(s) - V^\pi_{M'_r}(s) \geq 2\epsilon\}$ and $\{M' : \max_s V^\pi_{M'_r}(s) - V^\pi_{M_r}(s) \geq 2\epsilon\}$. The convexity of these sets depends on $(\pi, r, P)$ and, even if convex, may be disjoint. The following example illustrates this aspect.

**Example 4.6.** *Consider the MDP in Figure 1 with the target policy $\pi(s_1) = a_2$ and $\pi(s_2) = a_1$. The value functions are*

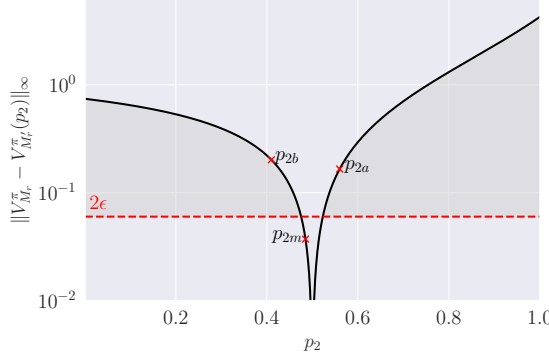

Figure 2. Plot of $\|V^\pi_M - V^\pi_{M'}(p_2)\|_\infty$ for varying values of $p_2$ in $M'$ (the other parameters are the same in both MDPs). The parameters $p_{2a} = 0.56$ and $p_{2b} = 0.41$ are both confusing parameters for $\epsilon = 0.03$ (above the red line), but their average $p_{2m}$ is not.

given by $V^\pi(s_2) = \theta V^\pi(s_1)$, where $\theta = \frac{\gamma p_3}{1-\gamma(1-p_3)}$, and $V^\pi(s_1) = \frac{p_2 r_2}{1-\gamma(p_2+(1-p_2)\theta)}$. Using parameters $\gamma = 0.9$, $r_2 = \frac{1}{2}$, $p_3 = 10^{-2}$, and $p_2 = \frac{1}{2}$ in the true model $M$, consider an alternative model $M'$ with the same $(p_1, r_1, r_2, \gamma)$. Varying $p_2$ in $M'$ shows that $p_2 = 0.56$ and $p_2 = 0.41$ are both confusing for $\epsilon = 0.03$, whereas their average is not. This phenomenon is illustrated in Figure 2.

In the next sub-section we explain how to circumvent this issue by considering a convex relaxation of the optimization problem in Eq. 2, denoted as the *relaxed characteristic time*.

## 4.3. Relaxed Characteristic Time

We proceed with finding a convex relaxation of $T_\epsilon(\omega; M)$ that holds for all distributions $\omega \in \Delta(\mathcal{S} \times \mathcal{A})$. Such relaxation not only upper bounds $T_\epsilon(\omega; M)$ in terms of $\mathcal{R}_\pi$ (instead of $\mathcal{R}_\pi^\epsilon$), but also allows us to better understand the scaling of $T_\epsilon^\star$. The proof can be found in Appendix B.2.3.

> **Theorem 4.7.** *For all $\omega \in \Delta(\mathcal{S} \times \mathcal{A})$ we have $T_\epsilon(\omega; M) \le U_\epsilon(\omega; M)$, where*
>
> $$U_\epsilon(\omega; M) := \sup_{\pi \in \Pi, r \in \mathcal{R}_\pi} \max_{s \in \mathcal{S}} \frac{\gamma^2 \|\rho_r^\pi(s)\|_\infty^2}{2\epsilon^2(1-\gamma)^2 \omega(s, \pi(s))}, \tag{6}$$
>
> *is convex in $\omega$. Let $U_\epsilon^\star(M) = U_\epsilon(\omega^\star; M)$ be the optimal rate, where $\omega^\star \in \arg\inf_{\omega \in \overline{\Omega(M)}} U_\epsilon(\omega; M)$ is an $U_\epsilon$-optimal allocation.*

This theorem exhibits some of the characteristics we mentioned before: as expected, the complexity is characterized by pairs $(s, s')$ for which the deviation $|\rho_r^\pi(s, s')|$ is large, for some worst-case policy-reward pair. What this result suggests is that sampling should be roughly proportional to the value deviation (a quantity that is a variance-like measure, as explained also in (Russo & Proutiere, 2023a)). However, quantifying the gap $|U_\epsilon(\omega; M) - T_\epsilon(\omega; M)|$ remains challenging, and we leave this analysis to future work.

**Scaling.** To better understand the scaling, for example, in the generative setting, choosing a uniform distribution $\omega(s, \pi(s)) = 1/S$ yields a scaling of $O\left(\max_{\pi, r} \frac{\gamma^2 |\mathcal{S}| \max_s \|\rho_r^\pi(s)\|^2}{2\epsilon^2(1-\gamma)^2}\right)$. Since $\|\rho_r^\pi(s)\|_\infty \le 1/(1-\gamma)$, we obtain a minimax scaling of $O\left(\frac{\gamma^2 |\mathcal{S}|}{2\epsilon^2(1-\gamma)^4}\right)$, which is independent of the number of policies or rewards. Note that this scaling can be misleading: because the sample complexity scales according to $\frac{\|\rho_r^\pi(s)\|_\infty^2}{\omega(s, \pi(s))}$, a small $\omega$ does not necessarily increase sample complexity if $\|\rho_r^\pi(s)\|_\infty^2$ is also small. In Appendix B.2.5 we depict the scaling on a simple example for two types of reward sets: reward-free (i.e., the entire set of rewards) and the circle $\|r\|_2 \le 1$.

### 4.3.1. OPTIMIZATION OVER CONVEX SETS OF REWARDS AND REWARD-FREE POLICY EVALUATION

For a convex reward set $\mathcal{R}_\pi$, the maximization over the rewards in Theorem 4.7 can be cast as a mixed-integer linear program (MILP). In practice, as shown in Appendix B.2.5, the same optimum is obtained more efficiently by solving a finite collection of ordinary convex programs.

In general, the solution is characterized by the matrix $\Gamma_{i,j}^\pi(s) = (K^\pi(s)G^\pi)_{i,j}$, where

$$K^\pi(s) = I - \mathbf{1}P(s, \pi(s))^\top, \quad G^\pi = (I - \gamma P^\pi)^{-1}.$$

$\Gamma_{i,j}^\pi(s)$ can be interpreted as the expected discounted number of visits to state $j$ starting from $i$ after subtracting the expected number of visits to $j$ starting from $s' \sim P(s, \pi(s))$. In other words, $\Gamma^\pi(s)$ is the analogue of the deviation $\rho$ in terms of discounted number of visits.

Then, for any convex reward set $\mathcal{R}_\pi$ the quantity $\sup_{r \in \mathcal{R}_\pi} |\rho_r^\pi(s, s')|$ is simply the larger of the optimal values of the two convex programs $\sup_{r \in \mathcal{R}_\pi} \pm e_{s'}^\top \Gamma^\pi(s)r$. Consequently, one only needs to solve a finite collection of standard convex problems to compute $U_\epsilon(\omega; M)$. We refer the reader to Appendix B.2.5 for details on how to solve $\inf_{\omega \in \Omega(M)} U_\epsilon(\omega; M)$,

**Reward-free scenario.** Lastly, in the reward-free scenario $\mathcal{R}_\pi = [0,1]^S, \forall \pi \in \Pi$, the optimization in $r$ admits a closed-form solution, which is determined by $\Gamma_+^\pi(s, s') := \sum_{j: \Gamma_{s',j}^\pi(s)>0} \Gamma_{s',j}^\pi(s)$, and $\Gamma_-^\pi(s, s') := -\sum_{j: \Gamma_{s',j}^\pi(s)<0} \Gamma_{s',j}^\pi(s)$.

> **Corollary 4.8.** *If $\mathcal{R}_\pi = [0,1]^S \; \forall \pi \in \Pi$, then*
>
> $$U_\epsilon(\omega; M) = \max_{\pi, s, s'} \frac{\gamma^2 \left[\max\left(\Gamma_+^\pi(s, s'), \Gamma_-^\pi(s, s')\right)\right]^2}{2\epsilon^2(1-\gamma)^2 \omega(s, \pi(s))}.$$

See also Appendix B.2.4 for a proof and Appendix B.2.5 for an example showing the reward-free sample complexity in the Riverswim environment (Strehl & Littman, 2004) for a single policy $\pi$. We are now ready to devise an algorithm based on the relaxed characteristic rate.

## 5. MR-NaS for Policy Evaluation

In this section we show how to adapt the **MR-NaS** (Multi-Reward Navigate and Stop) algorithm (Russo & Vannella, 2024) for multi-reward multi-policy evaluation based on the results from the previous section. MR-NaS (Algorithm 1) is a simple extension of NaS (Al Marjani et al., 2021), and is designed with 2 key components: (1) a sampling rule and (2) a stopping rule. We now discuss each of these.

**Sampling Rule.** The key idea is to sample according to the policy induced by $\omega^\star \in \arg\inf_{\omega \in \overline{\Omega(M)}} U_{\epsilon/2}(\omega; M)$. Indeed, sampling actions according to $\pi^\star(a|s) = \omega^\star(s,a)/\sum_b \omega^\star(s,b)$ guarantees optimality with respect to $U^\star_{\epsilon/2}$, as the solution $\omega^\star$ matches the relaxed rate in Theorem 4.4. The factor $\epsilon/2$ arises from the lower bound analysis that requires $2\epsilon$-separation. By tightening the accuracy to $\epsilon/2$, we ensure the $(\epsilon, \delta)$-PAC guarantee, which would otherwise be hard to prove. This results in an additional constant factor 4 in the sample complexity.

However, $\omega^\star$ cannot be computed without knowledge of the MDP $M$. As in previous works (Garivier & Kaufmann, 2016; Al Marjani et al., 2021), we employ the certainty equivalence principle (CEP): plug in the current estimate at time $t$ of the transition function and compute $\omega^\star_t$. The allocation $\omega^\star_t$ rapidly eliminates models confusing for $M_t$, efficiently determining whether the true model $M$ is non-confusing–motivating our use of the CEP.

To simplify the analysis, we make the following assumption:

**Assumption 5.1.** The solution $\omega^\star$ is unique and lies within the open set $\Omega(M)$.

Such assumption prevents: (1) the awkward case where a solution may belong to the boundary of $\overline{\Omega}(M)$; (2) multiple optimal solutions forming a convex set. This latter problem can be addressed using alternative techniques as in (Jedra & Proutiere, 2020; Russo et al., 2025; Russo & Proutiere, 2023b), which use regularization or the idea of tracking a convex combination of past solutions.

In summary, the algorithm proceeds as follows: at each time-step $t$ the agent computes the optimal visitation distribution $\omega^\star_t = \arg\inf_{\omega \in \overline{\Omega(M_t)}} U_{\epsilon/2}(\omega; M_t)$ with respect to $M_t$, the estimate of the MDP (which is, practically speaking, the estimate of the transition function). The policy $\pi^\star_t(a|s)$ induced by $\omega^\star_t$ is mixed with a *forcing* policy $\pi_{f,t}(\cdot|s)$ (e.g., a uniform distribution over actions or a distribution that encourages to select under-sampled actions; see also (Russo & Vannella, 2024) or Appendix C.3 for more details) that guar-

---

**Algorithm 1 MR-NaS** (Russo & Vannella, 2024)

1: **while** $t < U_{\epsilon/2}(N_t/t; M_t)\beta(N_t, \delta)$ **do**
2:     Compute $\omega^\star_t = \arg\inf_{\omega \in \overline{\Omega(M_t)}} U_{\epsilon/2}(\omega; M_t)$.
3:     Set $\pi_t(a|s_t) = (1 - \varepsilon_t)\pi^\star_t(a|s_t) + \varepsilon_t\pi_{f,t}(a|s_t)$,
    where $\pi^\star_t(a|s) = \omega^\star_t(s,a)/\sum_{a'} \omega^\star_t(s,a')$.
4:     Play $a_t \sim \pi_t(\cdot|s_t)$ and observe $s_{t+1} \sim P(\cdot|s_t, a_t)$.
5:     Update MDP estimate $M_t$ and set $t \leftarrow t + 1$.
6: **end while**

---

antees all actions are sampled infinitely often. The mixing factor $\varepsilon_t$ can be chosen as $\varepsilon_t = 1/\max(1, N_t(s_t))$, where $N_t(s)$ is the number of visits of state $s$ up to time $t$. The resulting exploration policy, $\pi_t$, is used to sample an action $a_t$, which, together with the communicating assumption and a forcing policy, yields an ergodic chain. Upon observing the next state, the transition function is updated using the empirical average.

**Stopping rule.** Lastly, the method stops whenever sufficient evidence has been gathered to obtain $(\epsilon, \delta)$-PAC guarantees. This requires approximately $U^\star_{\epsilon/2}(M)\log(1/\delta)$ samples (by inspecting Theorems 4.4 and 4.7).

This rule is defined by two quantities: (1) a threshold $\beta(N_t, \delta) = \log(1/\delta) + (S-1)\sum_{s,a}\log\left(e\left[1 + \frac{N_t(s,a)}{S-1}\right]\right)$; (2) the empirical characteristic time $U_{\epsilon/2}(N_t/t; M_t)$. In both, $N_t(s,a)$ is the number of times action $a$ has been selected in state $s$ up to time $t$, and $N_t = (N_t(s,a))_{s,a}$. In conclusion, we stop as soon as $t \geq U_{\epsilon/2}(N_t/t; M_t)\beta(N_t; \delta)$. Hence, we have the following guarantees (see proof in Appendix C.1).

> **Theorem 5.2.** *MR-NaS guarantees* $\mathbb{P}_M[\forall \pi \in \Pi, r \in \mathcal{R}_\pi : \|V^\pi_r - \hat{V}^\pi_r\|_\infty \leq \epsilon] \geq 1 - \delta$; $\mathbb{P}_M[\tau < \infty] = 1$ *and* $\limsup_{\delta \to 0} \frac{\mathbb{E}_M[\tau]}{\log(1/\delta)} \leq 4U^\star_\epsilon(M)$.

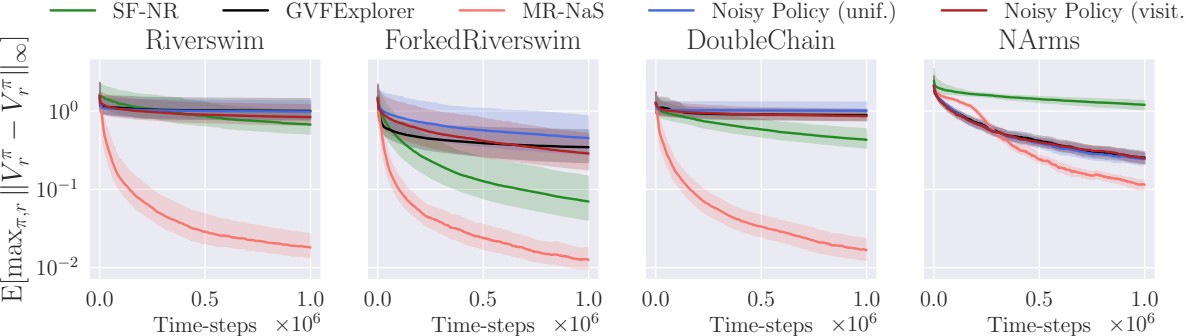

*Figure 3.* Multi-policy evaluation over finite set of rewards on different environments. Shaded curves represent 95% confidence intervals.

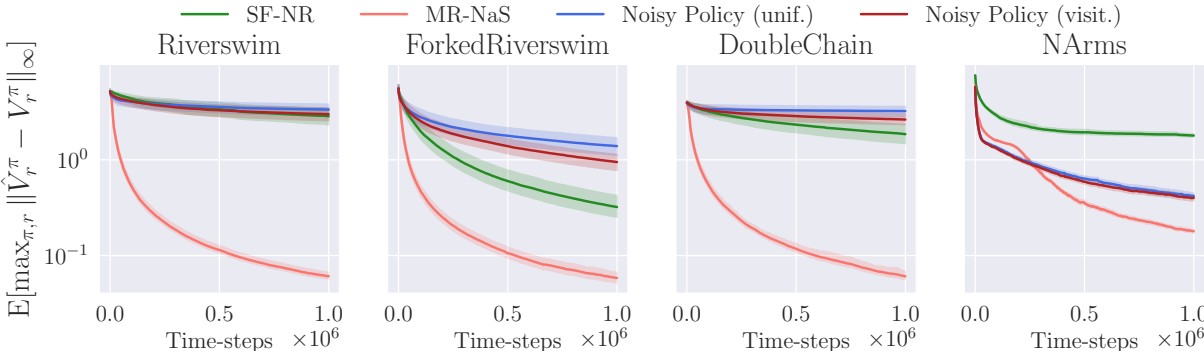

*Figure 4.* Reward-Free multi-policy evaluation. Here we depict the average error over the canonical basis $\mathcal{R}^{\pi}_{\mathrm{canon}}$ for each policy. Shaded curves represent 95% confidence intervals.

## 6. Numerical Results

In this section, we present the numerical results of **MR-NaS**, and other algorithms, on various environments with different reward sets (results use 30 seeds).

**Settings.** We study **MR-NaS** in 4 different environments: `Riverswim` (Strehl & Littman, 2004), `Forked Riverswim` (Russo & Proutiere, 2023a), `DoubleChain` (Kaufmann et al., 2021) and `NArms` (Strehl & Littman, 2004) (an adaptation of `SixArms`). For each environment we evaluated 3 scenarios: (1) multi-policy evaluation with finite reward sets for each policy; (2) reward-free multi-policy evaluation; (3) reward-free single policy evaluation (results for this one can be found in Appendix C.4). For each environment we evaluated 3 scenarios: (1) multi-policy evaluation with finite reward sets for each policy; (2) reward-free multi-policy evaluation; (3) reward-free single policy evaluation (results for this one can be found in Appendix C.4).

While our framework supports various settings, we focus on what we consider to be the most important and novel scenarios. For multi-policy evaluation, we sampled three random policies for each seed. These policies were sampled uniformly from the set of policies optimal for one-hot rewards, where each reward equals 1 at a single state-action pair and 0 elsewhere. In the case of finite reward sets, each policy was evaluated using the corresponding rewards from these sets. In the reward-free scenario, evaluations were conducted across the canonical basis $\mathcal{R}^{\pi}_{\mathrm{canon}}$ for each $\pi$.

**Algorithms.** While our work is one of the first to study the reward-free evaluation problem, there are some prior works that study the multi-task policy evaluation. We consider (1) `SF-NR` (McLeod et al., 2021), an algorithm for multi-task policy evaluation based on the Successor Representation, and we adapted it to also consider the reward-free setting (see Appendix C for more details). Next, we consider (2) `GVFExplorer` (Jain et al., 2024), a variance-based exploration strategy for learning general value functions (Sutton et al., 2011) based on minimizing the MSE. However, such exploration strategy is not applicable to the reward-free setting. We also evaluated (3) `Noisy`

`Policy - Uniform`, a mixture of the target policies $\pi_{\mathrm{mix}}(a|s) = \frac{|\{\pi \in \Pi : \pi(s) = a\}|}{|\Pi|}$, mixed with a uniform policy $\pi_u$ with a constant mixing factor $\varepsilon_t = 0.3$. The resulting behavior policy is $\pi_b = (1 - \varepsilon_t)\pi_{\mathrm{mix}} + \varepsilon_t \pi_u$. Lastly, (4) `Noisy Policy - Visitation`, computes the same behavior policy as in (3) with a non-constant mixing factor $\epsilon_t = 1/N_t(s_t)$, which is based on the number of visits.

**Discussion.** The results for the first two settings are shown in Figures 3 and 4 (policy evaluation was performed using the MDP estimate $M_t$ at each time-step). **MR-NaS** achieves good accuracy on all environments. On the other hand, `SF-NR` and `GVFExplorer` have mixed performance. While `SF-NR` is not designed to optimize a behavior policy, we note that the exploration strategy used by `GVFExplorer` is similar to solving a problem akin to Equation (2), but neglects the forward equations when optimizing the behavior policy (see also Appendix C.3 for details). As a result, `GVFExplorer` tends to perform worse in environments where certain rewards are hard to obtain under a uniform policy. Lastly, we refer the reader to Appendix C.4 for more details, and to the `README.md` file in the supplementary material to reproduce the results.

## 7. Conclusions

In this work, we studied the problem of devising an exploration strategy for online multi-reward multi-policy evaluation, accommodating reward sets that are either finite or convex, potentially encompassing all possible rewards. Leveraging tools from Best Policy Identification, we derived an instance-dependent sample complexity lower bound for the $(\epsilon, \delta)$-PAC setting. Based on this bound, we extended **MR-NaS** (Russo & Vannella, 2024) to the policy-evaluation setting, and showed its asymptotic efficiency. Lastly, we compared **MR-NaS** against other adaptive exploration methods across various domains, demonstrating its efficiency.

## Acknowledgments

The authors are pleased to acknowledge that the computational work reported on in this paper was performed on the Shared Computing Cluster which is administered by Boston University's Research Computing Services.

The authors would also like to thank Yilei Chen for the insightful discussions and constructive feedback provided during the drafting of this manuscript.

## Impact Statement

This paper presents work whose goal is to advance the field of Machine Learning and Reinforcement Learning. There are many potential societal consequences of our work, none which we feel must be specifically highlighted here.

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

# (Appendix)
# Adaptive Exploration for Multi-Reward Policy Evaluation

## Table of Contents

# A. Additional Notation

In this appendix we introduce some additional notation that is used in the proofs.

*Table 1.* Table of Notation

| Symbol | Definition | Description |
|---|---|---|
| $\mathrm{Alt}^\epsilon_{\pi,r}(M)$ | $\mathrm{Alt}^\epsilon_{\pi,r}(M) = \{M'_r : M \ll M'_r,\ \|V^\pi_{M_r} - V^\pi_{M'_r}\|_\infty > 2\epsilon\}$ | Set of alternative models in $(\pi, r)$. |
| $\mathcal{R}_\pi$ | $\mathcal{R}_\pi \subseteq [0,1]^S$ | Reward space considered for policy $\pi$. |
| $\mathcal{R}^\epsilon_\pi$ | $\mathcal{R}^\epsilon_\pi = \{r \in \mathcal{R}_\pi : \mathrm{Alt}^\epsilon_{\pi,r}(M) \neq \emptyset\}$ | The subset of rewards of $\mathcal{R}_\pi$ for which the set of alternative models is not empty. |
| $\mathbf{1}$ | $\mathbf{1} = \begin{bmatrix} 1 & \dots & 1 \end{bmatrix}$ | all-ones vector. |
| $e_i$ | $e_i = \begin{bmatrix} 0 & \dots & \underbrace{1}_{\text{i-th position}} & 0 & \dots \end{bmatrix}$ | $i$-th element of the canonical basis. |
| $\mathrm{diag}(A)$ | $\mathrm{diag}(A) = \begin{bmatrix} A_{11} & \dots & A_{nn} \end{bmatrix}^\top$ | Diagonal of a square matrix $A \in \mathbb{R}^{n \times n}$. |
| $P(s,a)$ | $P(s,a) = \begin{bmatrix} P(s_1\|s,a) & \dots & P(s_S\|s,a) \end{bmatrix}^\top$ | Vector of transition probabilities in $(s,a)$. |
| $P^\pi_s$ | $P^\pi_s = (P(s'\|s,\pi(s)))_{s' \in \mathcal{S}}$ | Vector of transition probabilities under $\pi$ in $s$. |
| $P^\pi$ | $P^\pi_{s,s'} = P(s'\|s,\pi(s)), P^\pi \in \mathbb{R}^{S \times S}$ | Matrix of transition probabilities under $\pi$. |
| $G^\pi$ | $G^\pi = (I - \gamma P^\pi)^{-1}, G^\pi \in \mathbb{R}^{S \times S}$ | Discounted fundamental matrix. |
| $\rho^\pi_r(s,s')$ | $\rho^\pi_r(s,s') = V^\pi_r(s') - \mathbb{E}_{\hat{s} \sim P(s,\pi(s))}[V^\pi_r(\hat{s})]$ | One-step value deviation in $(s,s')$. |
| $\rho^\pi_r(s)$ | $\rho^\pi_r(s) = V^\pi_r - (P^\pi_s)^\top V^\pi_r \mathbf{1}, \rho^\pi_r(s) \in \mathbb{R}^S$ | Vector of one-step value deviations in $s$. |
| $\rho^\pi_r$ | $\rho^\pi_r(s) = \mathbf{1}(V^\pi_r)^\top - (P^\pi V^\pi_r)\mathbf{1}^\top, \rho^\pi_r \in \mathbb{R}^{S \times S}$ | Matrix of one-step value deviations. |
| $K^\pi(s)$ | $K^\pi(s) \coloneqq (I_S - \mathbf{1}P(s,\pi(s))^\top), K^\pi(s) \in \mathbb{R}^{S \times S}$ | Deviation weighting matrix in $s$. |
| $\mathrm{sp}(V)$ | $\max_s V(s) - \min_s V(s)$ | Span of a value function. |
| $\Gamma^\pi(s)$ | $\Gamma^\pi(s) = K^\pi(s)G^\pi$ | Matrix used to express $\rho^\pi_r(s,s') = e^\top_{s'}\Gamma^\pi(s)r$. |
| $\mathrm{KL}_{P\|P'}(s,a)$ | $\mathrm{KL}(P(\cdot \mid s,a) \| P'(\cdot \mid s,a))$ | KL between next-state kernels at $(s,a)$. |
| $\|q\|_{\mathrm{TV}}$ | $\frac{1}{2}\|q\|_1$ | Total variation norm for signed vectors $q$. |
| $d^\pi(s)$ | $d^\pi = d^\pi P^\pi, \sum_s d^\pi(s) = 1$ | Stationary state distribution under $\pi$. |
| $d^\pi(s,a)$ | $d^\pi(s)\,\pi(a\|s)$ | Stationary state–action occupancy. |
| $\omega(s,a)$ | $\omega \in \Delta(\mathcal{S} \times \mathcal{A})$ | Stationary state–action allocation (design variable). |
| $N_t(s,a)$ | $\sum^t_{n=1} \mathbf{1}_{\{(s_n,a_n)=(s,a)\}}$ | Visit count to $(s,a)$ up to time $t$. |

# B. Theoretical Results

## B.1. Alternative Models and Value Deviation

In this section we prove some of the results in Section 4, and provide additional properties on the set of alternative models.

We begin by proving Proposition 4.1 and an additional necessary (minimax) condition for a model $M'$ to be an alternative model.

### B.1.1. PROOF OF PROPOSITION 4.1 AND AN ADDITIONAL MINIMAX NECESSARY CONDITION

*Proof of Proposition 4.1.* We first prove the sufficient condition, and then prove the necessary condition.

**Sufficient condition**. Let $s_0, s_1 \in \mathcal{S}$, and fix a reward $r \in \mathcal{R}$. In the following we omit the subscript $r$ for simplicity. Define a transition function $P'$ so that

$$P'(s'|s,a) = \begin{cases} P(s'|s,a) & s \neq s_0, \\ P(s'|s_0,a) & s = s_0 \wedge a \neq \pi(s_0), \\ \delta + (1-\delta)P(s_1|s_0,\pi(s_0)) & (s,a,s') = (s_0,\pi(s_0),s_1), \\ (1-\delta)P(s'|s_0,\pi(s_0)) & (s,a) = (s_0,\pi(s_0)) \wedge s' \neq s_1, \end{cases}$$

with $\delta \in (0,1)$. Therefore $P(s,a) \ll P'(s,a)$ for all $(s,a)$.

Let $V_P^\pi$ be the value of $\pi$ in $P$, and similarly define $V_{P'}^\pi$. For $s \neq s_0$ the difference in value $\Delta V^\pi = V_P^\pi - V_{P'}^\pi$ satisfies

$$\Delta V^\pi(s) = \gamma P^\pi(s)^\top \Delta V^\pi,$$

where $P^\pi(s) = \begin{bmatrix} P(s_1'|s,\pi(s)) & \dots & P(s_S'|s,\pi(s)) \end{bmatrix}^\top$ is the vector of transitions over the next state starting from $(s,\pi(s))$.

To analyse the case $s = s_0$ we indicate by $P'^\pi(s)$ the vector of transitions in $s$ for $P'$ under $\pi$. We obtain the following sequence of equalities

$$\begin{aligned} \Delta V^\pi(s_0) &= \gamma P^\pi(s_0)^\top V_P^\pi - \gamma \sum_{s'} P'(s'|s_0,\pi(s_0))V_{P'}^\pi(s'), \\ &= \gamma P^\pi(s_0)^\top V_P^\pi - \gamma(1-\delta)\sum_{s'\neq s_1} P(s'|s_0,\pi(s_0))V_{P'}^\pi(s') - \gamma\delta V_{P'}^\pi(s_1) - \gamma(1-\delta)P(s_1|s_0,\pi(s_0))V_{P'}^\pi(s_1), \\ &= \gamma P^\pi(s_0)^\top V_P^\pi - \gamma(1-\delta)\sum_{s'} P(s'|s_0,\pi(s_0))V_{P'}^\pi(s') - \gamma\delta V_{P'}^\pi(s_1), \\ &= \gamma[P^\pi(s_0)^\top(V_P^\pi - (1-\delta)V_{P'}^\pi) - \delta V_{P'}^\pi(s_1)], \\ &= \gamma[P^\pi(s_0)^\top(V_P^\pi - (1-\delta)V_{P'}^\pi) - \delta V_{P'}^\pi(s_1) \pm \delta P^\pi(s_0)^\top V_P^\pi \pm \delta V_P^\pi(s_1)], \\ &= \gamma[P^\pi(s_0)^\top(1-\delta)\Delta V^\pi + \delta\Delta V^\pi(s_1) + \delta(P^\pi(s_0)^\top V_P^\pi - V_P^\pi(s_1))], \\ &= \gamma P'^\pi(s_0)^\top \Delta V^\pi + \underbrace{\gamma\delta(P^\pi(s_0)^\top V_P^\pi - V_P^\pi(s_1))}_{=:b_{s_0}}. \end{aligned}$$

Hence, we can rewrite the expression of $\Delta V^\pi$ in matrix form as

$$\Delta V^\pi = b + \gamma P'^\pi \Delta V^\pi$$

where $b \in \mathbb{R}^S$, equal to $b_s = 0$ for $s \neq s_0$ and $b_{s_0} = \gamma\delta\left(\mathbb{E}_{s'\sim P(s_0,\pi(s_0))}[V_P^\pi(s')] - V_P^\pi(s_1)\right) = -\gamma\delta\rho_r^\pi(s_0,s_1)$.
Hence, we find

$$\begin{aligned} \|\Delta V^\pi\|_\infty &= \|(I - \gamma P'^\pi)^{-1}b\|_\infty, \\ &= |b_{s_0}| \cdot \left\|\sum_i (\gamma P'^\pi)^i e_{s_0}\right\|_\infty. \end{aligned}$$

Letting $M = \sum_i (\gamma P'^\pi)^i e_{s_0}$, observe that $M_{s_0} = e_{s_0}^\top M$ represents the (unnormalized) stationary discounted probability of reaching $s_0$ starting from $s_0$

$$M_{s_0} = \sum_i \gamma^i e_{s_0}^\top (P'^\pi)^i e_{s_0} = \sum_i \gamma^i \Pr(s_i = s_0 | s_0, \pi, P') \geq \sum_i \gamma^i P'^\pi (s_0 | s_0, \pi(s_0))^i = \frac{1}{1 - \gamma P'^\pi (s_0 | s_0, \pi(s_0))}.$$

Therefore

$$\|\Delta V^\pi\|_\infty \geq \frac{\gamma \delta |\rho_r^\pi(s_0, s_1)|}{1 - \gamma(1 - \delta) P(s_0 | s_0, \pi(s_0))}.$$

Let $p_{s_0}^\pi = P(s_0 | s_0, \pi(s_0))$. Then, we seek a value of $\delta \in (0, 1)$ so that

$$\frac{\gamma \delta |\rho_r^\pi(s_0, s_1)|}{1 - \gamma(1 - \delta) p_{s_0}^\pi} > 2\epsilon.$$

Some algebra gives that if $|\rho_r^\pi(s_0, s_1)| - 2\epsilon p_{s_0}^\pi > 0$ and $\gamma |\rho_r^\pi(s_0, s_1)| > 2\epsilon$ any value of $\delta$ in the range

$$\delta \in \left( \frac{2\epsilon(1 - \gamma p_{s_0}^\pi)}{\gamma(|\rho_r^\pi(s_0, s_1)| - 2\epsilon p_{s_0}^\pi)}, 1 \right)$$

leads to $\|\Delta V^\pi\|_\infty > 2\epsilon$, and thus the model is confusing. Since the above conditions can also be written as $|\rho_r^\pi(s_0, s_1)| > 2\epsilon \max(p_{s_0}^\pi, 1/\gamma) = 2\epsilon/\gamma$, and one can optimize in $s_1$, we find that a sufficient condition is that

$$\exists s_0 : \|\rho_r^\pi(s_0)\|_\infty > \frac{2\epsilon}{\gamma}.$$

**Necessary condition.** As before, let $V_M^\pi$ be the value of $\pi$ in $M$ with a reward $r$ (we omit the subscript $r$ for simplicity), and similarly define $V_{M'}^\pi$. If $M'$ is confusing for some reward $r \in \mathcal{R}$, then we have that there exists a state $s_0$ such that $2\epsilon < |\Delta V^\pi(s_0)|$, with $\Delta V^\pi(s) = V_M^\pi(s) - V_{M'}^\pi(s)$. Note that the following inequalities hold for any $s$

$$|\Delta V^\pi(s)| \leq \gamma \left| P(s, \pi(s))^\top V_M^\pi - P'(s, \pi(s))^\top V_{M'}^\pi \right|,$$
$$\leq \gamma \left| \Delta P(s, \pi(s))^\top V_M^\pi + P'(s, \pi(s))^\top \Delta V^\pi \right|,$$
$$\leq \gamma \left| \Delta P(s, \pi(s))^\top V_M^\pi \right| + \gamma \|\Delta V^\pi\|_\infty.$$

Since the inequality holds for all $s$, it implies that $2\epsilon < |\Delta V^\pi(s_0)| \leq \|\Delta V^\pi\|_\infty \leq \max_s \frac{\gamma}{1 - \gamma} |\Delta P(s, \pi(s))^\top V_M^\pi|$. We also denote by $s_{\max}$ the state maximizing this last quantity.

Letting $\mathbf{e}$ be the vector of ones, and noting that: (1) $\mathbb{E}_{s' \sim P(s, \pi(s))}[V_M^\pi(s')]$ is a constant and (2) $V_M^\pi - \mathbb{E}_{s' \sim P(s, \pi(s))}[V_M^\pi(s')]\mathbf{e} = \rho_r^\pi(s)$, then

$$2\epsilon < |\Delta V^\pi(s_0)| \leq \frac{\gamma}{1 - \gamma} \left| \Delta P(s_{\max}, \pi(s_{\max}))^\top \rho_r^\pi(s') \right|,$$
$$\leq \frac{2\gamma}{1 - \gamma} \|\Delta P(s_{\max}, \pi(s_{\max}))\|_{TV} \max_{s'} \|\rho_r^\pi(s')\|_\infty.$$

Since $\|\Delta P(s, \pi(s))\|_{TV} \leq 1$ for any $s$, we find that a necessary condition for $M'$ being a confusing model (for $r$) is that

$$\exists s : \|\rho_r^\pi(s)\|_\infty > \frac{\epsilon(1 - \gamma)}{\gamma}.$$

$\square$

In the following proof we provide an alternative minimax necessary condition for the existence of an alternative model for $(\pi, r)$. To derive the result, we use the following lemma.

---

**Lemma B.1** (Lemma 3 (Achiam et al., 2017)). *The divergence between discounted future state visitation distributions, $\|d^{\pi'} - d^\pi\|_1$, is bounded by an average divergence of the policies $\pi'$ and $\pi$:*

$$\|d^{\pi'} - d^\pi\|_1 \leq \frac{2\gamma}{1 - \gamma} \mathbb{E}_{s \sim d^\pi}[\|\bar{\pi}(s) - \bar{\pi}'(s)\|_{TV}]$$

---

Then, we have the following result.

> **Lemma B.2** (A necessary minimax condition for $M'$ to be a confusing model). *Consider a reward $r$ and a target policy $\pi$. If $M'$ is a confusing model, i.e. $||V_{M'}^\pi - V_M^\pi||_\infty > 2\epsilon$, then there $\exists s$ such that*
>
> $$\mathbb{E}_{a\sim\pi(\cdot|s)}[\text{KL}_{P|P'}(s,a)] > \frac{2(1-\gamma)^4}{\gamma^2}\epsilon^2.$$

*Proof.* The proof relies on constructing an alternative MDP and a policy $\bar\pi$ such that we can transform the problem of evaluating two different transition functions into a problem of comparing two different policies.

For simplicity, we also assume a deterministic policy $\pi$, and explain at the end how to extend the argument to a general stochastic policy.

**Imaginary MDP $\bar M$.** We begin by constructing an imaginary deterministic MDP $\overline{M}$ where the action space is $\bar{\mathcal{A}} = \mathcal{S}$. The transition function of this MDP is $\bar P(s'|s,u) = \mathbf{1}_{\{u=s'\}}$, i.e., taking the action $u$ leads the agent to state $s'$ with probability 1. The reward function instead is $\bar r(s) = r(s,\pi(s))$.

Define now the policy $\bar\pi(u|s) = P(u|s,\pi(s))$. With $\overline{M}_{\bar r}$, we can convert the value of the policy under the original MDP and confusing models to the value of different policies under $\overline{M}_{\bar r}$. Specifically, the value of the policy $\pi$ (deterministic) under the MDP $M_r$ with transitions $P$ is

$$V_{M_r}^\pi(s) = V_{\bar M_{\bar r}}^{\bar\pi}(s).$$

This follows from the fact that

$$V_{\bar M_{\bar r}}^{\bar\pi}(s) = \sum_{s',u} \bar\pi(u|s)\bar P(s'|s,u)[\bar r(s) + \gamma V_{\bar M_{\bar r}}^{\bar\pi}(s')] = \sum_{s'} P(s'|s,\pi(s))[r(s,\pi(s)) + \gamma V_{\bar M_{\bar r}}^{\bar\pi}(s')].$$

Then, by an appropriate application of the Bellman operator one can see that $V_{\bar M_{\bar r}}^{\bar\pi}(s) = V_{M_r}^\pi(s)$

**Bounding using the difference lemma.** Consider now a confusing model $M_r' \in \text{Alt}_{\pi,r}^\epsilon(M)$ with transition $P'$, and define a policy $\bar\pi'(u|s) = P'(u|s,\pi(s))$. Hence, we have

$$|V_{M_r}^\pi(s) - V_{M_r'}^\pi(s)| = |V_{\bar M_{\bar r}}^{\bar\pi}(s) - V_{\bar M_{\bar r}}^{\bar\pi'}(s)|. \tag{7}$$

Based on Lemma B.1 and Equation (7), we have $\forall s$

$$|V_{M_r}^\pi(s) - V_{M_r'}^\pi(s)| = |V_{\bar M_{\bar r}}^{\bar\pi}(s) - V_{\bar M_{\bar r}}^{\bar\pi'}(s)|, \tag{8}$$

$$\leq \frac{1}{1-\gamma}||d_{\bar M_{\bar r}}^{\bar\pi}(s) - d_{\bar M_{\bar r}}(s)^{\bar\pi'}||_1, \tag{9}$$

$$\leq \frac{2\gamma}{(1-\gamma)^2}\mathbb{E}_{s'\sim d_{\bar M_{\bar r}}^{\bar\pi}(s)}[||\bar\pi(s) - \bar\pi'(s)||_{TV}], \tag{10}$$

where $d_{\bar M_{\bar r}}^{\bar\pi}(s)$ denotes the visitation distribution induced by $\bar\pi$ in $\bar M_{\bar r}$ starting from $s$.

If $\forall s$, we have $||\bar\pi(s) - \bar\pi'(s)||_{TV} \leq \frac{(1-\gamma)^2}{\gamma}\epsilon$, then $|V_{M_r}^\pi(s) - V_{M_r'}^\pi(s)| \leq 2\epsilon, \forall s$, i.e. $M_r'$ is not a confusing model. Hence, there exists $s$ such that

$$\frac{(1-\gamma)^2}{\gamma}\epsilon \leq ||\bar\pi(s) - \bar\pi'(s)||_{TV} = ||P(s,\pi(s)) - P'(s,\pi(s))||_{TV} \leq \sqrt{\frac{1}{2}\text{KL}_{P|P'}(s,\pi(s))}.$$

**Extension to a stochastic policy.** The extension to a stochastic policy involves a few more steps, and it is omitted for simplicity. It follows from defining an imaginary MDP $\bar M$ with $\bar P(s'|s,u) = \mathbf{1}_{\{u=s'\}}$, $\bar r(s,u) = \sum_{a'} r(s,a')\pi(a'|s)$ and $\bar\pi(u|s) = \sum_a P(u|s,a)\pi(a|s)$. The argument concludes by noting that if $M$ is confusing then

$$\frac{(1-\gamma)^4}{\gamma^2}\epsilon^2 \leq ||\bar\pi(s) - \bar\pi'(s)||_{TV}^2 \leq \frac{1}{2}\text{KL}(\bar\pi(s),\bar\pi'(s)) \leq \frac{1}{2}\mathbb{E}_{a\sim\pi(\cdot|s)}\left[\text{KL}_{P|P'}(s,a)\right].$$

by the data-processing inequality. $\qquad\square$

### B.1.2. ADDITIONAL RESULTS ON THE VALUE DEVIATION $\rho$

**Value deviation.** To analyze these confusing sets, and their implications for sample complexity, we define the following instance-dependent quantity, that we refer to as the *one-step value deviation*:

$$\rho_r^\pi(s, a, s') := V_r^\pi(s') - \mathbb{E}_{\hat{s}\sim P(s,a)}[V_r^\pi(\hat{s})] \quad \forall s, s' \in \mathcal{S},$$
$$\rho_r^\pi(s, s') := \rho_r^\pi(s, \pi(s), s').$$

This quantity measures how much the value at $s'$ differs from the expected value of the next state when starting at $s$.

We also define these quantities in vector form $\rho_r^\pi(s) := \begin{bmatrix} \rho_r^\pi(s, s_1) & \cdots & \rho_r^\pi(s, s_S) \end{bmatrix}^\top$, so that $\|\rho_r^\pi(s)\|_\infty = \max_{s'} |\rho_r^\pi(s, s')|$ is the maximum one-step deviation at $s$ (similarly, one defines $\rho_r^\pi(s, a)$). The deviation $\rho_r^\pi$ is closely related to the *span* of the value function

$$\text{sp}(V_r^\pi) := \max_{s'} V_r^\pi(s') - \min_s V_r^\pi(s), \tag{11}$$

but, is in general smaller. In the following lemma we also show that it is unlikely that $\max_{s,s'} \rho_r^\pi(s, s')$ is achieved for $s' = s$, but rather for $s' \neq s$. Depending on the characteristics of the MDP, it is more plausible that $\rho_r^\pi(s, s) \approx 0$.

---

**Lemma B.3.** *For any reward vector $r \in [0, 1]^S$, states $s, s' \in \mathcal{S}$, we have: (I) $|\rho_r^\pi(s, s)| \leq 1$; (II) $|\rho_r^\pi(s, s')| \leq \text{sp}(V_r^\pi)$ and (III)*
$$|\rho_r^\pi(s, s') - \rho_r^\pi(s', s)| \leq |\Delta_r^\pi(s, s')| + \Gamma_{s,s'}^\pi \text{sp}(V_r^\pi),$$
*where $\Delta_r^\pi(s, s') := r(s, \pi(s)) - r(s', \pi(s'))$ and $\Gamma_{s,s'}^\pi := \frac{1+\gamma}{2}\|P(s, \pi(s)) - P(s', \pi(s'))\|_1$.*

---

*Proof of Lemma B.3.* **First part.** For the first part of the lemma, note that

$$\rho_r^\pi(s, s) = V_r^\pi(s) - P(s, \pi(s))^\top V_r^\pi = r(s, \pi(s)) + (\gamma - 1)P(s, \pi(s))^\top V_r^\pi.$$

Since $P(s, \pi(s))^\top V_r^\pi \geq 0$, and $\gamma \in (0, 1)$, then $(\gamma - 1)P(s, \pi(s))^\top V_r^\pi \leq 0$. Using that the reward is bounded in $[0, 1]$, we obtain $\rho_r^\pi(s, s) \leq 1$.

Then, using that $0 \leq V_r^\pi(s) \leq 1/(1 - \gamma) \Rightarrow -1 \leq (\gamma - 1)V_r^\pi(s)$ we also have $\rho_r^\pi(s, s) \geq r(s, \pi(s)) - 1 \geq -1$. Thus $|\rho_r^\pi(s, s)| \leq 1$.

**Second part.** The second part is rather straightforward, and follows from the definition of span $\text{sp}(V_r^\pi) = \max_s V_r^\pi(s) - \min_s V_r\pi(s)$. Indeed we have

$$\rho_r^\pi(s, s') = V_r^\pi(s') - P(s, \pi(s))^\top V_r^\pi \leq \max_s V_r^\pi(s) - \min_s V_r^\pi(s) = \text{sp}(V_r^\pi)$$

and

$$\rho_r^\pi(s, s') \geq \min_s V_r^\pi(s) - \max_s P(s, \pi(s))^\top V_r^\pi \geq -\text{sp}(V_r^\pi).$$

**Third part.** For the third and last part we first note the rewriting

$$\rho_r^\pi(s, s') - \rho_r^\pi(s', s) = V_r^\pi(s') - V_r^\pi(s) + [P(s', \pi(s')) - P(s, \pi(s))]^\top V_r^\pi,$$
$$= r(s', \pi(s')) - r(s, \pi(s)) + (\gamma + 1)[P(s', \pi(s')) - P(s, \pi(s))]^\top V_r^\pi,$$

Then, define $\mathcal{Z}(s, s') := \{z \in \mathcal{S} : P(z|s', \pi(s')) \geq P(z|s, \pi(s))\}$ and

$$P_+^\pi(s, s') := \sum_{z \in \mathcal{Z}(s,s')} P(z|s', \pi(s')) - P(z|s, \pi(s)), \quad P_-^\pi(s, s') := \sum_{z \in \mathcal{S}\setminus\mathcal{Z}(s,s')} P(z|s', \pi(s')) - P(z|s, \pi(s)).$$

Also observe $P_+^\pi(s, s') = -P_-^\pi(s, s')$, and $\|P(s, \pi(s)) - P(s', \pi(s'))\|_1 = P_+^\pi(s, s') - P_-^\pi(s, s')$, from which follows that $P_+^\pi(s, s') = \frac{1}{2}\|P(s, \pi(s)) - P(s', \pi(s'))\|_1$. Then, we have that

$$
\begin{aligned}
[P(s', \pi(s')) - P(s, \pi(s))]^\top V_r^\pi &= \sum_z [P(z|s', \pi(s')) - P(z|s, \pi(s))] V_r^\pi(z), \\
&= \sum_{z \in \mathcal{Z}(s, s')} [P(z|s', \pi(s')) - P(z|s, \pi(s))] V_r^\pi(z) \\
&\quad + \sum_{z \in \mathcal{S} \setminus \mathcal{Z}(s, s')} [P(z|s', \pi(s')) - P(z|s, \pi(s))] V_r^\pi(z), \\
&\leq P_+^\pi(s, s') \max_s V_r^\pi(s) + \sum_{z \in \mathcal{S} \setminus \mathcal{Z}(s, s')} \underbrace{[P(z|s', \pi(s')) - P(z|s, \pi(s))]}_{<0} V_r^\pi(z), \\
&\leq P_+^\pi(s, s') \max_s V_r^\pi(s) + \sum_{z \in \mathcal{S} \setminus \mathcal{Z}(s, s')} [P(z|s', \pi(s')) - P(z|s, \pi(s))] \min_s V_r^\pi(s), \\
&\leq P_+^\pi(s, s') \max_s V_r^\pi(s) + \underbrace{P_-^\pi(s, s')}_{=-P_+^\pi(s, s')} \min_s V_r^\pi(s), \\
&\leq P_+^\pi(s, s') \mathrm{sp}(V_r^\pi), \\
&\leq \frac{1}{2}\|P(s, \pi(s)) - P(s', \pi(s'))\|_1 \mathrm{sp}(V_r^\pi).
\end{aligned}
$$

By taking the absolute value one obtains the result. $\qquad\square$

Another useful metric is the dispersion factor $\lambda_r^\pi$, which is defined as

$$
\lambda_r^\pi := \frac{\min_s V_r^\pi(s)}{\max_s V_r^\pi(s)},
$$

from which we see its relationship with the span, i.e., $\mathrm{sp}(V_r^\pi) / \max_s V_r^\pi(s) = 1 - \lambda_r^\pi$.

The parameter $\lambda_r^\pi$ measures the spread of the value function across different states. Specifically, a smaller $\lambda_r^\pi$ indicates a greater dispersion of values among states.

---

**Lemma B.4.** *The dispersion factor $\lambda_r^\pi$ satisfies:*

*1. $V_r^\pi(s') \geq \lambda_r^\pi V_r^\pi(s)$ for any pair $s, s' \in \mathcal{S}$.*

*2. $\max_s V_r^\pi(s) - V_r^\pi(s') \leq (1 - \lambda_r^\pi) \max_s V_r^\pi(s)$ for any state $s'$.*

---

*Proof.* First, clearly $\lambda_r^\pi \in [0, 1]$. Then, the first property is derived from the following inequalities that hold for any pair $(s, s')$:

$$
\lambda_r^\pi V_r^\pi(s) \leq \lambda_r^\pi \max_s V_r^\pi(s) = \min_s V_r^\pi(s) \leq V_r^\pi(s').
$$

The second statement stems from the simple fact that $-V_r^\pi(s') \leq -\min_s V_r^\pi(s)$ for any $s'$. Hence:

$$
\max_s V_r^\pi(s) - V_r^\pi(s') \leq \max_s V_r^\pi(s) - \min_s V_r^\pi(s) = \max_s V_r^\pi(s) - \lambda_r^\pi \max_s V_r^\pi(s).
$$

$\qquad\square$

Using this metric, we are able to provide the following result, providing a bound on $\rho_r^\pi(s, s')$ (note that one could rewrite the following result in terms of the span).

---

**Lemma B.5.** *For any reward vector $r \in [0, 1]^S$, we have:$\rho_r^\pi(s, s') \in \left[ -\frac{1 - \lambda_r^\pi \gamma}{1 - \gamma}, \max\left(1, \frac{1 - \lambda_r^\pi}{1 - \gamma}\right) \right]$ for all $s, s' \in \mathcal{S}$. Furthermore, for $\lambda_r^\pi = \gamma$ we have that $|\rho_r^\pi(s, s')| \leq 2$ for all $s, s'$.*

---

*Proof.* **First part.** For the first part of the lemma, note that

$$\rho_r^\pi(s,s) = V_r^\pi(s) - P(s,\pi(s))^\top V_r^\pi = r(s,\pi(s)) + (\gamma-1)P(s,\pi(s))^\top V_r^\pi.$$

Since $P(s,\pi(s))^\top V_r^\pi \geq 0$, and $\gamma \in (0,1)$, then $(\gamma-1)P(s,\pi(s))^\top V_r^\pi \leq 0$. Using that the reward is bounded in $[0,1]$, we obtain $\rho_r^\pi(s,s) \leq 1$.

Then, using that $0 \leq V_r^\pi(s) \leq 1/(1-\gamma) \Rightarrow -1 \leq (\gamma-1)V_r^\pi(s)$ we also have $\rho_r^\pi(s,s) \geq r(s,\pi(s)) - 1 \geq -1$. Thus $|\rho_r^\pi(s,s)| \leq 1$.

**Second part.** For the second part of the lemma let $P_s^\pi = P(s,\pi(s))$ be the vector of transition probabilities in $s$ under action $\pi(s)$. We first prove that $V_r^\pi(s) \geq \lambda_r^\pi V_r^\pi(s') \Rightarrow (P_s^\pi)^\top V_r^\pi \geq \lambda_r^\pi (P_{s'}^\pi)^\top V_r^\pi$. The proof is simple, and follows from the following inequalities that hold for any pair $(s,\hat{s}) \in \mathcal{S}^2$:

$$(P_s^\pi)^\top V_r^\pi \geq \min_{s'} V_r^\pi(s') \geq \lambda_r^\pi \max_{s'} V_r^\pi(s') \geq \lambda_r^\pi (P_{\hat{s}}^\pi)^\top V_r^\pi.$$

We can use this fact to prove the result. From the proof of Proposition 4.2 we know that

$$\rho_r^\pi = \mathbf{1}(V_r^\pi)^\top - (P^\pi V_r^\pi)\mathbf{1}^\top = \mathbf{1}(r + \gamma P_\pi V_r^\pi)^\top - (P^\pi V_r^\pi)\mathbf{1}^\top,$$

where $r$ is the vector of rewards. Let $W = P_\pi V_r^\pi$ and note the two facts (1) $W_s = (P_s^\pi)^\top V_r^\pi$ and (2) $0 \leq W_s \leq 1/(1-\gamma)$. Then, one can see that $(\rho_r^\pi)_{s,s'}$ is

$$(\rho_r^\pi)_{s,s'} = \rho_r^\pi(s,s') = r(s') + \gamma W_{s'} - W_s.$$

We prove the statement in two steps:

1. We first prove $\rho_r^\pi(s,s') \leq \max\left(1, \frac{1-\lambda_r^\pi}{1-\gamma}\right)$. Using that for any pair $(s,s')$ we have $\lambda_r^\pi W_{s'} \leq W_s$ it follows that

$$
\begin{aligned}
r(s') + \gamma W_{s'} - W_s &= r(s') + \gamma W_{s'} - W_s \pm \lambda_r^\pi W_{s'}, \\
&= r(s') + (\gamma - \lambda_r^\pi)W_{s'} + \underbrace{\lambda_r^\pi W_{s'} - W_s}_{\leq 0}, \\
&\leq r(s') + (\gamma - \lambda_r^\pi)W_{s'}, \\
&\leq 1 + \max\left(0, \frac{\gamma - \lambda_r^\pi}{1-\gamma}\right), \\
&\leq \max\left(1, \frac{1-\lambda_r^\pi}{1-\gamma}\right).
\end{aligned}
$$

2. We now prove a lower bound on $\rho_r^\pi(s,s')$. The idea is to seek a value of $\eta \geq 0$ such that $\gamma W_{s'} - W_s \geq (\gamma-1)W_s\eta$, which implies $\gamma W_{s'} - W_s \geq -\eta$ (using that $-W_s \geq -1/(1-\gamma)$). Then, we find $W_{s'} \geq \frac{(\eta\gamma-\eta+1)}{\gamma}W_s$. Since $W_{s'} \geq \lambda_r^\pi W_s$, we can set

$$\frac{(\eta\gamma-\eta+1)}{\gamma} = \lambda_r^\pi \Rightarrow \eta = \frac{\lambda_r^\pi\gamma - 1}{\gamma - 1},$$

(notably, if $\lambda_r^\pi = \gamma$, then $\eta = 1 + \gamma \leq 2$). Lastly, we obtain $\rho_r^\pi(s,s') \geq r(s') + \gamma W_{s'} - W_s \geq -\eta = -\frac{1-\lambda_r^\pi\gamma}{1-\gamma}$, which concludes the proof.

$\square$

Hence, if the value is similar across states, we can expect a smaller sample complexity.

B.1.3. PROOF OF PROPOSITION 4.2 AND RELATED RESULTS

In the following, we prove Proposition 4.2, which characterizes the set of rewards for which $\max_s \|\rho_r^\pi(s)\|_\infty = 0$. In the proof we define the following deviation matrix

$$\rho_r^\pi = \begin{bmatrix} \rho_r^\pi(s_1) & \cdots & \rho_r^\pi(s_S) \end{bmatrix}^\top, \tag{12}$$

that is used to prove the proposition.

*Proof of Proposition 4.2.* First, note that

$$\rho_r^\pi(s, s') = V_{M_r}^\pi(s') - \mathbb{E}_{\hat{s} \sim P(s, \pi(s))}[V_r^\pi(\hat{s})],$$
$$= (e_{s'} - P_s)^\top V_r^\pi,$$

where $e_s$ is the $s$-th element of the canonical basis, $P_s = P(s, \pi(s))$. Hence, we can write $\rho_r^\pi(s)$ as

$$\rho_r^\pi(s) = V_r^\pi - P_s^\top V_r^\pi \mathbf{1}$$

where $\mathbf{1}$ is the vector of ones. Therefore

$$\rho_r^\pi = \mathbf{1}(V_r^\pi)^\top - (P^\pi V_r^\pi)\mathbf{1}^\top,$$

where $P^\pi$ is a $S \times S$ matrix satisfying $(P^\pi)_{s, s'} = P(s'|s, \pi(s))$ (and $P_s^\pi = (P(s'|s, \pi(s)))_{s' \in \mathcal{S}}$). We consider the condition $\rho_r^\pi = 0$ being identically 0, that is

$$\mathbf{1}(V_r^\pi)^\top = (P^\pi V_r^\pi)\mathbf{1}^\top \iff V_r^\pi = (P_s^\pi V_r^\pi)\mathbf{1} \quad \forall s \in \mathcal{S}.$$

Since $(P_s^\pi V_r^\pi)\mathbf{1} = \mathbf{1}(P_s^\pi)^\top V_r^\pi$, letting $M_s^\pi = \mathbf{1}(P_s^\pi)^\top$, then $\rho_r^\pi = 0 \iff V_r^\pi = M_s^\pi V_r^\pi$ for every $s \in \mathcal{S}$, that is, $V_r^\pi$ is a right eigenvector of $M_s^\pi$ with eigenvalue 1 for all $s \in \mathcal{S}$. Since $M_s^\pi$ has rank 1 for every $s$, then $V_r^\pi$ is the only right eigenvector for eigenvalue 1 for all states.

Now, one can easily see that $M_s^\pi \mathbf{1} = \mathbf{1}(P_s^\pi)^\top \mathbf{1} = \mathbf{1}$, thus the right eigenvector of $M_s^\pi$ associated to the eigenvalue 1 is the ones vector $\mathbf{1}$ itself, for all states.

To translate this condition onto a condition on the rewards, we use the fact that $V_r^\pi = G^\pi r$, with $G^\pi = (I - \gamma P^\pi)^{-1}$ (which is invertible). Therefore, we require $r$ to satisfy

$$\exists c \in \mathbb{R} : c\mathbf{1} = G^\pi r \iff c(I - \gamma P^\pi)\mathbf{1} = r.$$

**Conclusion.** Using that $P^\pi \mathbf{1} = \mathbf{1}$, we conclude that

$$\rho_r^\pi = 0 \iff r \propto \mathbf{1}.$$

The set of all vectors $r$ for which $\rho_r^\pi = 0$ is precisely the set $\{\alpha\mathbf{1} : \alpha \in \mathbb{R}\}$. Thus, $r$ must be a scalar multiple of $\mathbf{1}$. $\qquad\square$

We conclude this subsection by characterizing in what cases, $\rho_r^\pi(s, s)$ is identically zero across states. First, note the following result.

> **Lemma B.6.** *Let* $\mathrm{diag}(\rho_r^\pi) = \begin{bmatrix} \rho_r^\pi(s_1, s_1) & \dots & \rho_r^\pi(s_S, s_S) \end{bmatrix}^\top$. *We have that*
>
> $$\mathrm{diag}(\rho_r^\pi) = (I - P^\pi)(I - \gamma P^\pi)^{-1}r. \tag{13}$$

*Proof.* First, note that

$$\rho_r^\pi(s, s') = V_{M_r}^\pi(s') - \mathbb{E}_{\hat{s} \sim P(s, \pi(s))}[V_{M_r}^\pi(\hat{s})],$$
$$= (e_{s'} - P_s)^\top V_{M_r}^\pi,$$
$$= (e_{s'} - P_s)^\top G^\pi r,$$

where $e_s$ is the $s$-th element of the canonical basis, $P_s = P(s, \pi(s))$ and in the last step we used the fact that $V_{M_r}^\pi = G^\pi r$ with $G^\pi = (I - \gamma P^\pi)^{-1}$.

Since for any $s$ we can write $\rho_r^\pi(s, s) = (e_s - P_s)^\top G^\pi r$, it follows that $\rho_r^\pi(s, s) = [(I - P^\pi)G^\pi r]_s$, and thus $\mathrm{diag}(\rho_r^\pi) = (I - P^\pi)G^\pi r$. $\qquad\square$

We now consider a result that permits us to understand when $\max_s \rho_r^\pi(s, s) = 0$ for any state $s$, which is the diagonal of $\rho_r^\pi$. In the result, we consider a partition of the set of recurrent states of $P^\pi$ into $m$ disjoint closed irreducible sets $C_1, \dots, C_m$. Then, define $\tau_i = \inf\{t : s_t \in C_i\}$ be the hitting time of $C_i$, that is, the earliest time $C_i$ is reached, and define

$h_i^\pi(s) = \mathbb{P}^\pi(\tau_i < \infty | s)$ to be the probability of reaching $C_i$ in finite time starting from state $s$ under $\pi$. We have then the following result.

---

**Proposition B.7.** *Let $m^\pi$ be the number of disjoint closed irreducible sets $C_1, \ldots, C_{m^\pi}$ in $P^\pi$. Define the vector $h_i^\pi \in \mathbb{R}^S$ as $(h_i^\pi)_s := \mathbb{P}^\pi(\tau_i < \infty | s)$, which is the probability of reaching $C_i$ in finite time starting from $s$ under $\pi$. Then, for any non-zero reward vector $r \in \mathbb{R}^S$, we have that:*

$$\max_s |\rho_r^\pi(s, s)| = 0 \iff r \in \mathrm{span}\{h_1^\pi, \ldots, h_{m^\pi}^\pi\}.$$

*In other words, the reward $r$ lies in the span of the $m^\pi$ vectors $(h_i^\pi)_{i=1}^{m^\pi}$.*

---

In other words, the one-step value deviation at state $s$ is zero provided that the reward function is constant on each irreducible closed set of $P^\pi$

*Proof.* From Lemma B.6, we know that

$$\mathrm{diag}(\rho_r^\pi) = (I - P^\pi)(I - \gamma P^\pi)^{-1} r.$$

Then, let $x = (I - \gamma P^\pi)^{-1} r$ and consider the solutions to $(I - P^\pi)x = 0$. In other words, we are interested in the real-valued right eigenvectors of $P^\pi$ associated to the eigenvalue 1, that is, $P^\pi x = x$. Then, for any such right eigenvector we obtain

$$(I - \gamma P^\pi)x = r \Rightarrow (1 - \gamma)x = r.$$

Hence, since $x$ is a right eigenvector ($P^\pi x = x$), $r$ is also a right-eigenvector of $P^\pi$ associated to its eigenvalue 1 (note that we get $P^\pi r = r$).

Therefore, finding the reward vectors satisfying $\|\mathrm{diag}(\rho_r^\pi)\|_\infty = 0$ amounts to finding the real-valued right eigenvectors of $P^\pi$ associated to its eigenvalue 1.

In general, for an irreducible stochastic transition matrix $P^\pi$ it is well-known (Seneta, 2006) that the only real-valued eigenvector associated to the eigenvalue 1 is the all-ones vector, and thus $r_\lambda = \lambda \mathbf{1}$ is a solution for any $\lambda \in \mathbb{R}$.

For a generic $P^\pi$, the above argument can be extended by partitioning the set of recurrent states into $m$ disjoint closed irreducible sets $C_1, \ldots, C_m$. Therefore, as shown in (Puterman, 2014), the state space can be rewritten as $\mathcal{S} = C_1 \cup C_2 \cup \ldots C_m \cup T$, where $T$ is the set of transient states. After relabeling the states, we can express the transition matrix $P^\pi$ as

$$P^\pi = \begin{bmatrix} P_1 & 0 & \ldots & 0 & 0 \\ 0 & P_2 & \ldots & 0 & 0 \\ \vdots & & \ddots & & \vdots \\ 0 & 0 & \ldots & P_m & 0 \\ Q_1 & Q_2 & \ldots & Q_m & Q_{m+1} \end{bmatrix}$$

where $P_i$ corresponds to the transition function in $C_i$, $Q_i$ to transitions from states in $T$ to states in $C_i$, and $Q_{m+1}$ to transition between states in $T$ (Puterman, 2014). Moreover, we also know from (Puterman, 2014, Theorem A.5) that the eigenvalue 1 has multiplicity $m$.

Note that, since each $P_i$ is an irreducible stochastic matrix, there is only one right eigenvector associated to 1, and it is the vectors of ones, of size $|C_i|$. Then, if $Q_i = 0$, then one can conclude that $(h_i)_s = \mathbf{1}_{(s \in C_i)}$ is a right eigenvector of $P^\pi$. However, that is not generally the case. Therefore we now derive a general way to find the right eigenvectors.

Define $\tau_i = \inf\{t : s_t \in C_i\}$ be the hitting time of $C_i$, that is, the earliest time $C_i$ is reached, and define $h_i^\pi(s) = \mathbb{P}^\pi(\tau_i < \infty | s)$, that is, the probability of reaching $C_i$ in finite time starting from state $s$ under $\pi$.

Then, we observe that $h_i^\pi(s)$ satisfies a recursive relationship

$$
\begin{aligned}
h_i^\pi(s) &= \mathbb{P}^\pi(\tau_i < \infty | s_1 = s), \\
&= \sum_{s'} \mathbb{P}^\pi(\tau_i < \infty, s_2 = s' | s_1 = s), \\
&= \sum_{s'} P^\pi(s'|s) \mathbb{P}^\pi(\tau_i < \infty | s_2 = s'), \\
&= \sum_{s'} P^\pi(s'|s) h_i^\pi(s'),
\end{aligned}
$$

where we used the strong Markov property. Therefore $h_i^\pi = P^\pi h_i^\pi$, meaning that $h_i^\pi$ is a right eigenvector of $P^\pi$ associated to the eigenvalue 1. Now, observe that $h_i^\pi(s) = 1$ for all $s \in C_i$, and $h_i^\pi(s) = 0$ for all $s \in C_j, j \neq i$. Because of that, these vectors are distinct eigenvectors of the eigenvalue 1. Since the eigenvalue 1 has algebraic (and geometric) multiplicity $m$ by Theorem A.5 in (Puterman, 2014), hence the $m$ independent vectors $(h_i^\pi)_{i=1}^m$ form a basis of the eigenspace and therefore span it.

In conclusion, we have that $\{h_1^\pi, \ldots, h_m^\pi\}$ are $m$ linearly independent eigenvectors of $P^\pi$ associated to the eigenvalue 1. Thus, any $r \in \text{span}\{h_1^\pi, \ldots, h_m^\pi\}$ yields $\|\text{diag}(\rho_r^\pi)\|_\infty = 0$. $\qquad\square$

## B.2. Sample Complexity Results

Now we provide proofs for the sample complexity lower bound, and related results.

### B.2.1. PROOF OF THEOREM 4.4

*Proof of Theorem 4.4.* The overall proof strategy is to derive a set of constraints on the number of visits $N_t(s,a) = \sum_{n=1}^t \mathbf{1}_{((s_n,a_n)=(s,a))}$ to a state-action pair $(s,a)$ that any $(\epsilon, \delta)$-PAC strategy must guarantee. Dividing $\mathbb{E}_M[N_t(s,a)]$ by $\mathbb{E}_M[\tau]$ yields a quantity $\omega(s,a)$ that is a distribution over the state-action space, and as $\delta \to 0$, $\omega$ converges to a stationary distribution. In the last part of the proof we discuss the ergodicity of the chain induced by this stationary distribution.

**Part 1: set of constraints through a change of measure.** The initial part of the proof follows the same technique as in (Al Marjani et al., 2021; Russo & Proutiere, 2023a), and leverages change of measure arguments (Lai & Robbins, 1985; Garivier & Kaufmann, 2016).

Consider a policy-reward pair $\pi \in \Pi, r \in \mathcal{R}_\pi$. For a confusing model $M'_{\pi,r} \in \text{Alt}_{\pi,r}^\epsilon(M)$ consider the log-likelihood ratio up to time $t$ of a sequence of observations $(z_1, z_2, \ldots)$, with $z_i = (s_i, a_i)$, under the original MDP $M_r$ and an alternative model $M'_{\pi,r} \in \text{Alt}_{\pi,r}^\epsilon(M)$:

$$
L_{t,\pi,r} = \sum_{n=1}^t \log \frac{P(s_{n+1}|z_n)}{P'_{\pi,r}(s_{n+1}|z_n)}.
$$

Then, as in (Al Marjani & Proutiere, 2021), one can show that for all $t \in \mathbb{N}$ the following equality holds

$$
\begin{aligned}
\mathbb{E}_M[L_{t,\pi,r}] &= \sum_{s,a} \sum_{u=1}^\infty \mathbb{P}_M[N_t(s,a) \geq u] \text{KL}_{P|P'_{\pi,r}}(s,a), \\
&= \sum_{s,a} \mathbb{E}_M[N_t(s,a)] \text{KL}_{P|P'_{\pi,r}}(s,a).
\end{aligned}
$$

Now we proceed to lower bound the expected log-likelihood ratio at the stopping time $\tau$. We indicate by $\mathbb{P}_{M'_{\pi,r}}$ the measure induced by $M'_{\pi,r}$. Applying the information processing inequality in (Kaufmann et al., 2016), we lower bound the expected log-likelihood at the stopping time $\tau$ as

$$
\mathbb{E}_M[L_{\tau,\pi,r}] \geq \text{kl}(\mathbb{P}_M(\mathcal{E}), \mathbb{P}_{M'_{\pi,r}}(\mathcal{E}))
$$

for some event $\mathcal{E}$ that is $\mathcal{F}_\tau$-measurable. Then, define the event $\mathcal{E} = \{\exists \pi \in \Pi, r \in \mathcal{R}_\pi : \|V_{M_r}^\pi - \hat{V}_r\|_\infty > \epsilon\}$. By definition of $(\epsilon, \delta)$-PAC algorithm we have $\mathbb{P}_M(\mathcal{E}) \leq \delta$. If we can show that $\mathbb{P}_{M'_{\pi,r}}(\mathcal{E}) \geq 1 - \delta$, then we can lower bound $\text{kl}(\mathbb{P}_M(\mathcal{E}), \mathbb{P}_{M'_{\pi,r}}(\mathcal{E})) \geq \text{kl}(\delta, 1 - \delta)$ by the monotonicity properties of the KL-divergence.

To show $\mathbb{P}_{M'_{\pi,r}}(\mathcal{E}) \geq 1 - \delta$, a few steps are needed:

$$
\begin{aligned}
\mathbb{P}_{M'_{\pi,r}}(\mathcal{E}) &\overset{(a)}{\geq} \max_{\bar{\pi}\in\Pi, \bar{r}\in\mathcal{R}_{\bar{\pi}}} \mathbb{P}_{M'_{\pi,r}}(\|V^{\bar{\pi}}_{M_{\bar{r}}} - \hat{V}_{\bar{r}}\|_{\infty} > \epsilon), \\
&\overset{(b)}{\geq} \max_{\bar{\pi}\in\Pi, \bar{r}\in\mathcal{R}_{\bar{\pi}}} \mathbb{P}_{M'_{\pi,r}}(\|V^{\bar{\pi}}_{M_{\bar{r}}} - V^{\bar{\pi}}_{M'_{\bar{r}}}\|_{\infty} - \|V^{\bar{\pi}}_{M'_{\bar{r}}} - \hat{V}_{\bar{r}}\|_{\infty} > \epsilon), \\
&\geq \mathbb{P}_{M'_{\pi,r}}(\|V^{\pi}_{M_r} - V^{\pi}_{M'_r}\|_{\infty} - \|V^{\pi}_{M'_r} - \hat{V}_r\|_{\infty} > \epsilon), \\
&\overset{(c)}{\geq} \mathbb{P}_{M'_{\pi,r}}(\|V^{\pi}_{M'_r} - \hat{V}_r\|_{\infty} < \epsilon), \\
&\geq \min_{\pi\in\Pi, r\in\mathcal{R}_{\pi}} \mathbb{P}_{M'_{\pi,r}}(\|V^{\pi}_{M'_r} - \hat{V}_r\|_{\infty} < \epsilon), \\
&\overset{(d)}{\geq} \mathbb{P}_{M'_{\pi,r}}(\forall \pi\in\Pi, r\in\mathcal{R}_{\pi} : \|V^{\pi}_{M'_r} - \hat{V}_r\|_{\infty} < \epsilon) \\
&= 1 - \mathbb{P}_{M'_{\pi,r}}(\exists \pi\in\Pi, r\in\mathcal{R}_{\pi} : \|V^{\pi}_{M'_r} - \hat{V}_r\|_{\infty} \geq \epsilon) \geq 1 - \delta.
\end{aligned}
$$

where (a) and (d) follow from the Fréchet inequalities; (b) is an application of the triangle inequality; (c) follows from the fact that $M'_{\pi,r}$ is confusing for $(\pi, r)$ and the last inequality from the definition of $(\epsilon, \delta)$-PAC algorithm. Henceforth, for a $(\epsilon, \delta)$-PAC algorithm one can conclude that $\mathbb{E}_M[L_{\tau,\pi,r}] \geq \mathrm{kl}(\delta, 1 - \delta)$.

As the inequality above holds for all $\pi \in \Pi, r \in \mathcal{R}_{\pi}$, we obtain the following set of constraints:

$$
\inf_{\pi\in\Pi, r\in\mathcal{R}_{\pi}, M'_r\in\mathrm{Alt}^{\epsilon}_{\pi,r}(M)} \sum_{s,a} \mathbb{E}_M[N_{\tau}(s,a)]\mathrm{KL}_{P|P'_r}(s,a) \geq \mathrm{kl}(\delta, 1 - \delta).
$$

**Part 2: optimizing the stationary distribution.**

Now, let $n(s,a) = \mathbb{E}_M[N_{\tau}(s,a)]$ and $\omega(s,a) = \mathbb{E}_M[N_{\tau}(s,a)]/\mathbb{E}_M[\tau]$. We rewrite the constraints above as follows

$$
\mathbb{E}_M[\tau] \inf_{\pi\in\Pi, r\in\mathcal{R}_{\pi}, M'_r\in\mathrm{Alt}^{\epsilon}_{\pi,r}(M)} \sum_{s,a} \omega(s,a)\mathrm{KL}_{P|P'_r}(s,a) \geq \mathrm{kl}(\delta, 1 - \delta).
$$

Hence, the optimization problem revolves around optimizing $\omega$ in $\inf_{\pi\in\Pi, r\in\mathcal{R}_{\pi}, M'_r\in\mathrm{Alt}^{\epsilon}_{\pi,r}(M)} \sum_{s,a} \omega(s,a)\mathrm{KL}_{P|P'_r}(s,a)$. By Lemma 1 in (Al Marjani et al., 2021) we also know that an algorithm that navigates the MDP needs to satisfy the constraints

$$
\left| \sum_a n(s,a) - \sum_{s',a'} P(s|s',a')n(s',a') \right| \leq 1 \quad \forall s \in \mathcal{S}.
$$

After normalizing $n(s,a)$ by $n(s,a)/\log(1/\delta)$, and using that $\mathrm{kl}(\delta, 1-\delta) \underset{\delta\to 0}{\sim} \log(1/\delta)$, as $\delta \to 0$ we find the following lower bound

$$
\liminf_{\delta\to 0} \frac{\mathbb{E}_M[\tau]}{\log(1/\delta)} \geq \frac{1}{H^{\star}_{\epsilon}(M)},
$$

where $H^{\star}$ formulates the optimization problem over $\omega$ as follows

$$
\begin{aligned}
H^{\star}_{\epsilon}(M) &:= \sup_{\omega\in\Delta(\mathcal{S}\times\mathcal{A})} \inf_{\pi\in\Pi, r\in\mathcal{R}_{\pi}, M'_r\in\mathrm{Alt}^{\epsilon}_{\pi,r}(M)} \sum_{s,a} \omega(s,a)\mathrm{KL}_{P|P'_r}(s,a), \\
&\text{subject to } \forall s\in\mathcal{S}, \sum_a \omega(s,a) = \sum_{s',a'} P(s|s',a')\omega(s',a').
\end{aligned}
$$

**Part 3: irreducibility of the stationary distribution.** Let $\Omega_0(M) = \{\omega \in \Delta(\mathcal{S}\times\mathcal{A}) : \sum_a \omega(s,a) = \sum_{s',a'} P(s|s',a')\omega(s',a') \ \forall s \in \mathcal{S}\}$. We note that a solution $\omega^{\star} \in \Omega_0(M)$ to the above problem that characterizes $H^{\star}_{\epsilon}(M)$ may not necessarily induce an irreducible chain, i.e., under the induced transition $P_{\omega^{\star}}(s'|s) = \frac{\sum_a P(s'|s,a)\omega^{\star}(s,a)}{\sum_b \omega^{\star}(s,b)}$ some states may not be accessible ($s'$ is accessible from $s$ if there exists $n \in \mathbb{N}_+$ such that $\mathbb{P}_M(s_{t+n} = s'|s_t = s, \omega^{\star}) > 0$

for all $t$ under $\omega^\star$ (Puterman, 2014)). Consider an arbitrary starting state $s_0$ (by Assumption 3.1). If a state $s_c$ is not accessible from $s_0$ under $\omega^\star$, then the number of visits to that state is bounded, i.e., $\sup_{t\geq 1}\sum_a \mathbb{E}_M[N_t(s_c, a)] < \infty$.

Under Assumption 4.3 there exists $\pi \in \Pi, r \in \mathcal{R}_\pi$ such that we can build a confusing model $M'_r$. In particular, we can build a confusing model $M'_r$ as in the proof of Proposition 4.1 (see the "sufficient" part of the proof), satisfying

$$
\begin{aligned}
\mathrm{kl}(1-\delta, \delta) &\leq \sum_{s,a} \mathbb{E}_M[N_\tau(s,a)]\mathrm{KL}_{P|P'_r}(s,a), \\
&= \mathbb{E}_M[N_\tau(s_c, \pi(s_c))]\mathrm{KL}_{P|P'_r}(s_c, \pi(s_c)), \\
&= \mathbb{E}_M[N_\tau(s_c, \pi(s_c))]\Big[(1 - P(s_1|s_c, \pi(s_c)))\log\left(\frac{1}{1-\lambda}\right) \\
&\qquad + P(s_1|s_c, \pi(s_c))\log\left(\frac{P(s_1|s_c, \pi(s_c))}{\lambda + (1-\lambda)P(s_1|s_c, \pi(s_c))}\right)\Big], \\
&< \infty.
\end{aligned}
$$

where $s_1 \in \arg\max_{s\in\mathcal{S}}|\rho_r^\pi(s_c, s)|$ and $\lambda$ is a fixed value in $(0,1)$, independent of $\delta$, taken as in the proof of Proposition 4.1 (in that proof we use $\delta$ instead of $\lambda$, not to be confused with the confidence parameter). The boundedness of the right-hand side comes from the boundedness of $\sum_a \mathbb{E}_M^\pi[N_t(s_c, a)] < \infty$ for all $t \geq 1$ and that $\lambda$ is a fixed value in $(0,1)$. However, as $\delta \to 0$ the left-hand side scales as $\log(1/\delta) \to \infty$, which violates the constraint (and the constraint should hold for all $\pi \in \Pi, r \in \mathcal{R}_\pi$). Since this holds for any pair $(s_0, s_c)$, and $s_0$ is arbitrary, an optimal solution $\omega^\star$ must induce an irreducible chain over $\mathcal{S}$.

*Remark: in general, note that the set of confusing models puts constraints on pairs $(s, \pi(s))_{s\in\mathcal{S}, \pi\in\Pi}$, but not necessarily on other state-action pairs. Hence, without further assumptions, we do not have guarantees that an optimal solution yields an irreducible chain. This is in contrast to previous work (Al Marjani et al., 2021; Russo & Proutiere, 2023a; Russo & Vannella, 2024), where constraints also affect pairs $s, a \neq \pi(s)$.*

**Last step: ergodicity.** This last step, while not necessary, shows that any optimal irreducible allocation can be well approximated by a solution $\omega$ satisfying $\omega(s, a) > 0$ for all $(s, a)$. We show that any $\omega$ satisfying this latter property forms a dense set in $\Omega_0(M)$ (and, since such $\omega(s, a)$ is also irreducible over $\mathcal{S}$, it's also dense in the set $\Omega_0(M) \cap \{\omega : P_\omega$ is irreducible over $\mathcal{S}\}$).

Consider any $\omega^\star \in \Omega_0(M)$. Let $\lambda \in (0,1)$ and $\omega' \in \Omega(M)$. Define then $\omega_\lambda = (1-\lambda)\omega^\star + \lambda\omega'$.

Then, for any convergent sequence $(\lambda_k)_k$ such that $\lim_{k\to\infty}\lambda_k = 0$, we have that

$$\|\omega^\star - \omega_{\lambda_k}\| = \lambda_k\|\omega^\star - \omega'\| \to 0.$$

Since for every $k$ we have that $\omega_{\lambda_k} \in \Omega(M)$, and limit points are in $\Omega_0(M)$, it follows that $\Omega(M)$ is dense in $\Omega_0(M)$.

$\square$

### B.2.2. OPTIMALITY OF BEHAVIOR POLICIES

An interesting question is to understand in which cases a policy is optimal for exploration. For instance, consider the MDP in Figure 1 and the single target policy $\pi(\cdot|s) = a_2 \ \forall s$. For that MDP, it is *in general* sub-optimal to sample according to such policy, since under $\pi$ state $s_1$ becomes transient. In the following lemma we prove that a necessary condition for an optimal exploration policy is to guarantee that states with large deviation gap are visited infinitely often.

> **Lemma B.8.** *Let $T^{\pi_e}$ be the set of transient states under an exploration policy $\pi_e$. Assume that $T^{\pi_e} \neq \emptyset$. If there exists $s_c \in T^{\pi_e}, \pi \in \Pi, r \in \mathcal{R}_\pi$ such that $\|\rho_r^\pi(s_c)\|_\infty > 2\epsilon/\gamma$, then $\pi_e$ is not optimal, in the sense that*
>
> $$T_\epsilon(d^{\pi_e}; M)^{-1} = 0, \tag{14}$$
>
> *for any stationary distribution $d^{\pi_e}$ (of the chain $P^{\pi_e}$) induced by $\pi_e$.*

*Proof of Lemma 4.5.* Since $s_c$, by assumption, is transient under $\pi_e$, the expected number of returns of the Markov chain to state $s_c$ is bounded, i.e., $\sup_{t\geq 1}\sum_a \mathbb{E}_M^{\pi_e}[N_t(s_c, a)] < \infty$. In other words, we have that $d^{\pi_e}(s_c) = 0$.

Now, the condition $\|\rho_r^\pi(s_c)\|_\infty > 2\epsilon/\gamma$ guarantees that there exists a confusing model by Proposition 4.1. Let $s_1 \in \arg\max_s |\rho_r^\pi(s_c, s_1)|$. Then, one can define an alternative model $P'$ as in the proof of Proposition 4.1:

$$
P'(s'|s,a) = \begin{cases}
P(s'|s,a) & s \neq s_c, \\
P(s'|s_c,a) & s = s_c \wedge a \neq \pi(s_c), \\
\delta + (1-\delta)P(s_1|s_c,\pi(s_c)) & (s,a,s') = (s_c,\pi(s_c),s_1), \\
(1-\delta)P(s'|s_c,\pi(s_c)) & (s,a) = (s_c,\pi(s_c)) \wedge s' \neq s_1.
\end{cases}
$$

Then, from the proof of Proposition 4.1 we know that there exists $\delta \in (0,1)$ such that $P'$ is an alternative model.

Hence, for any stationary distribution $d^{\pi_e}$ induced by $\pi_e$ we obtain

$$
\sum_{s,a} d^{\pi_e}(s,a)\mathrm{KL}(P(s,a), P'(s,a)) = \sum_a d^{\pi_e}(s_c,a)\mathrm{KL}(P(s_c,a), P'(s_c,a)).
$$

where $d^{\pi_e}(s,a) = d^{\pi_e}(s)\pi_e(a|s)$ (and $d^{\pi_e}(s) = \sum_a d^{\pi_e}(s,a)$). But $0 = d^{\pi_e}(s_c) = \sum_a d^{\pi_e}(s_c,a)$. Which, by non-negativity of $d^{\pi_e}$, implies that $d^{\pi_e}(s_c,a) = 0$ for every $a$. Hence

$$
\sum_{s,a} d^{\pi_e}(s,a)\mathrm{KL}(P(s,a), P'(s,a)) = 0.
$$

Since $P'$ is an alternative confusing model, we have that

$$
\inf_{M' \in \mathrm{Alt}_{\pi,r}^\epsilon(M)} \sum_{s,a} d^{\pi_e}(s,a)\mathrm{KL}_{P|P'}(s,a) = 0,
$$

and thus

$$
(T_\epsilon(\omega^{\pi_e}; M))^{-1} = 0 \quad \text{for any stationary } \omega^{\pi_e}(s,a) = d^{\pi_e}(s)\omega(s,a)
$$

$\square$

Therefore, one concludes that a necessary condition for an optimal exploration strategy is to guarantee that states with large deviation gap are visited infinitely often.

We conclude this section with a sufficient condition on the optimality of a general behavior policy $\pi_\beta$, when used to evaluate a single policy $\pi$ on a single reward. What the proposition points out is that $\pi_\beta$ needs to induce a sampling distribution $d^{\pi_\beta}$ with large sampling rates where $\|\rho_r^\pi(s)\|_\infty$ is sufficiently large. The extension to multi-reward follows naturally.

---

**Proposition B.9.** *Consider a behavior policy $\pi_\beta$ and a target policy $\pi$. Assume that $\pi_\beta$ induces an irreducible class over $\mathcal{S}$ (i.e., there are no transient states). Denote by $d^{\pi_\beta}$ the stationary distribution induced by $\pi_\beta$, and assume that $\pi_\beta$ satisfies $d^{\pi_\beta}(s,\pi(s)) > 0$ for all $s$.*
*Then, let $\mathcal{S}_{\mathrm{cnf}} = \{s \in \mathcal{S} : \|\rho_r^\pi(s)\|_\infty > 3\epsilon/\gamma\}$ be the set of confusing states and assume that $\mathcal{S}_{\mathrm{cnf}} \neq \emptyset$. Define $\varepsilon^{\pi,\pi_\beta}(s) := \max(0, \sup_{\omega \in \Omega(M)} \omega(s) - d^{\pi_\beta}(s,\pi(s)))$ and $\Delta^{\pi,\pi_\beta}(s) = d^{\pi_\beta}(s,\pi(s)) - \min_{s'} d^{\pi_\beta}(s',\pi(s'))$.*
*Then, under the assumption that $\epsilon \geq \gamma/(1-\gamma)^2$, we have that the behavioral policy $\pi_\beta$ is $2\lambda^{\pi,\pi_\beta}$-optimal in the following sense*

$$
\inf_{M' \in \mathrm{Alt}_{\pi,r}^\epsilon(M)} \sum_{s,a} d^{\pi_\beta}(s,a)\mathrm{KL}_{P|P'}(s,a) \geq \sup_{\omega \in \Omega(M)} \inf_{M' \in \mathrm{Alt}_{\pi,r}^\epsilon(M)} \sum_{s,a} \omega(s,a)\mathrm{KL}_{P|P'}(s,a) - 2\lambda^{\pi,\pi_\beta},
$$

*where $\lambda^{\pi,\pi_\beta} = \min_{s \in \mathcal{S}_{\mathrm{cnf}}}(\varepsilon^{\pi,\pi_\beta}(s) + \Delta^{\pi,\pi_\beta}(s))$.*

---

*Proof.* First, note that the first assumption $\mathcal{S}_{\mathrm{cnf}} \neq \emptyset$ implies that the set of alternative models $\mathrm{Alt}_{r,\pi}^\epsilon$ is not empty by Proposition 4.1.

Let $s_0 \in \mathcal{S}_{\mathrm{cnf}}$. Consider an alternative model $P'$ similar to the one used in the proof of Proposition 4.1: $P'(s'|s,a) = P(s'|s,a)$ for all $s \neq s_0, a, s'$, and $P'(s_0|s_0,a) = \delta + P(s_0|s_0,a)(1-\delta)$, $P'(s'|s_0,a) = (1-\delta)P(s'|s_0,a)$ for $s' \neq s_0$, with $\delta \in (0,1)$. From the same proposition we also know that if $\delta \in (0,1)$ satisfies

$$
\frac{\gamma\delta\|\rho_r^\pi(s_0)\|_\infty}{1-\gamma(1-\delta)p_{s_0}^\pi} > 2\epsilon,
$$

then the model is confusing (where $p_{s_0}^\pi = P(s_0|s_0, \pi(s_0))$). Since $\frac{\gamma\delta\|\rho_r^\pi(s_0)\|_\infty}{1-\gamma(1-\delta)p_{s_0}^\pi} \geq \gamma\delta\|\rho_r^\pi(s_0)\|_\infty$, a simpler way to guarantee that $P'$ is an alternative model is to choose $\delta$ satisfying $\delta > \frac{2\epsilon}{\gamma\|\rho_r^\pi(s_0)\|_\infty}$. Setting $\delta = 2/3$, we find

$$\|\rho_r^\pi(s_0)\|_\infty > \frac{3\epsilon}{\gamma},$$

which is true by assumption. Hence, for $\delta = 2/3$ the model is confusing.

We now proceed with an upper bound of the KL divergence between $P$ and $P'$ in $s_0$:

$$\begin{aligned}
\mathrm{KL}_{P|P'}(s_0, a) &= \sum_{s'} P(s'|s_0, a) \log\left(\frac{P(s'|s_0, a)}{P'(s'|s_0, a)}\right), \\
&\leq \log(\frac{1}{1-\delta}), \\
&\leq \frac{\delta}{1-\delta}, \\
&\leq 2.
\end{aligned}$$

The first inequality holds by plugging in the definition of $P'$ and noticing $\log(\frac{P(s_0|s_0,a)}{\delta+(1-\delta)P(s_0|s_0,a)}) \leq \log(\frac{P(s_0|s_0,a)}{(1-\delta)P(s_0|s_0,a)})$; the second inequality holds due to $\log(x) \leq x - 1$; the last inequality uses that $\delta \leq 2/3$ implies $\delta/(1-\delta) \leq 2$.

Define $\varepsilon^{\pi,\pi_\beta}(s) := \max(0, \sup_{\omega\in\Omega(M)} \omega(s) - d^{\pi_\beta}(s, \pi(s)))$. Then

$$\begin{aligned}
\inf_{M'\in\mathrm{Alt}_{\pi,r}^\epsilon(M)} \sum_{s,a} \omega(s,a)\mathrm{KL}_{P|P'}(s,a) &\leq \sum_{s,a} \omega(s,a)\mathrm{KL}_{P|P'}(s,a), \\
&= \sum_a \omega(s_0,a)\mathrm{KL}_{P|P'}(s_0,a), \\
&\leq 2\omega(s_0) \\
&\leq 2\omega(s_0) \pm 2d^{\pi_\beta}(s_0, \pi(s_0)), \\
&\leq 2d^{\pi_\beta}(s_0, \pi(s_0)) + 2[\sup_{\omega\in\Omega} \omega(s_0) - d^{\pi_\beta}(s_0, \pi(s_0))], \\
&\leq 2d^{\pi_\beta}(s_0, \pi(s_0)) + 2\varepsilon^{\pi,\pi_\beta}(s_0).
\end{aligned}$$

where we used $\omega(s) := \sum_a \omega(s,a)$. On the other hand, we have shown above that for a model to be a confusing model, the necessary condition is $\exists \hat{s}$ such that $KL_{P|P'}(\hat{s}, \pi(\hat{s})) \geq \frac{2(1-\gamma)^4}{\gamma^2}\epsilon^2$ (see Lemma B.2). Hence, defining $\Delta^{\pi,\pi_\beta}(s) = d^{\pi_\beta}(s, \pi(s)) - \min_{s'} d^{\pi_\beta}(s', \pi(s'))$, we have,

$$\begin{aligned}
\inf_{M'\in\mathrm{Alt}_{\pi,r}^\epsilon(M)} \sum_{s,a} d^{\pi_\beta}(s,a)\mathrm{KL}_{P|P'}(s,a) &\geq \inf_{M'\in\mathrm{Alt}_{\pi,r}^\epsilon(M)} \sum_a d^{\pi_\beta}(\hat{s},a)\mathrm{KL}_{P|P'}(\hat{s},a), \\
&\geq \inf_{M'\in\mathrm{Alt}_{\pi,r}^\epsilon(M)} d^{\pi_\beta}(\hat{s})\pi_\beta(\pi(\hat{s})|\hat{s})\mathrm{KL}_{P|P'}(\hat{s}, \pi(\hat{s})), \\
&= \inf_{M'\in\mathrm{Alt}_{\pi,r}^\epsilon(M)} d^{\pi_\beta}(\hat{s}, \pi(\hat{s}))\mathrm{KL}_{P|P'}(\hat{s}, \pi(\hat{s})), \\
&\geq d^{\pi_\beta}(\hat{s}, \pi(\hat{s}))\frac{2(1-\gamma)^4}{\gamma^2}\epsilon^2, \\
&\geq 2[d^{\pi_\beta}(\hat{s}, \pi(\hat{s})) \pm d^{\pi_\beta}(s_0, \pi(s_0))], \\
&\geq 2d^{\pi_\beta}(s_0, \pi(s_0)) + 2[d^{\pi_\beta}(\hat{s}, \pi(\hat{s})) - d^{\pi_\beta}(s_0, \pi(s_0))], \\
&\geq 2d^{\pi_\beta}(s_0, \pi(s_0)) + 2[\min_{s'} d^{\pi_\beta}(s', \pi(s')) - d^{\pi_\beta}(s_0, \pi(s_0))], \\
&\geq 2d^{\pi_\beta}(s_0, \pi(s_0)) - 2\Delta^{\pi,\pi_\beta}(s_0).
\end{aligned}$$

where we used in the first equality that $d^{\pi_\beta}(s,a) = d^{\pi_\beta}(s)\pi_\beta(a|s)$, and in the third inequality that $\frac{(1-\gamma)^4}{\gamma^2}\epsilon^2 \geq 1$.

Conclusively, we have shown that for all $\omega \in \Omega(M), s_0 \in \mathcal{S}_{\mathrm{cnf}}$

$$\inf_{M' \in \mathrm{Alt}_{\pi,r}^{\epsilon}(M)} \sum_{s,a} d^{\pi_\beta}(s,a) \mathrm{KL}_{P|P'}(s,a) \geq \inf_{M' \in \mathrm{Alt}_{\pi,r}^{\epsilon}(M)} \sum_{s,a} \omega(s,a) \mathrm{KL}_{P|P'}(s,a) - 2(\Delta^{\pi,\pi_\beta}(s_0) + \varepsilon^{\pi,\pi_\beta}(s_0)).$$

Therefore

$$\inf_{M' \in \mathrm{Alt}_{\pi,r}^{\epsilon}(M)} \sum_{s,a} d^{\pi_\beta}(s,a) \mathrm{KL}_{P|P'}(s,a) \geq \inf_{M' \in \mathrm{Alt}_{\pi,r}^{\epsilon}(M)} \sum_{s,a} \omega(s,a) \mathrm{KL}_{P|P'}(s,a) - 2\lambda^{\pi,\pi_\beta} \qquad \forall \omega \in \Omega(M).$$

$\square$

The previous results, with the following one, are useful when studying how mixing a target policy with a uniform policy affects sample complexity. In the following lemma we study the rate of visits at stationarity when the target policy is mixed with a uniform distribution. Let $P_u$ be the transition function induced by the uniform policy, i.e., $P_u(s'|s) = \sum_a P(s'|s,a)/A$, and let $P_u^k$ be the $k$-th step transition matrix induced by the uniform policy. From the ergodicity of $P_u$ there exists an integer $k_0$ such that $P_u^k(s'|s) > 0$ for all $k \geq k_0$ and all $(s',s) \in \mathcal{S}^2$ (the existence of $k_0$ is guaranteed by Levin & Peres (2017, Proposition 1.7)).

Hence, to that end, define

$$\eta_k = \min_{s,s'} P_u^k(s'|s),$$

be the minimal probability of reaching $s'$ from $s$ in $k$ steps, and let $k_0 = \min\{k \in \mathbb{N} : \eta_k > 0\}$. We then have the following result.

> **Lemma B.10.** *Let $\pi = (1-\epsilon)\pi_{tgt} + \epsilon\pi_u$ be a mixture policy defined as the mixture between a target policy $\pi_{tgt}$ and a uniform policy $\pi_u$, with mixing coefficient $\epsilon \in (0,1]$. Let $d^\pi(s)$ denote the average number of visits to state $s$ under policy $\pi$ at stationarity. Then, for all states we have*
>
> $$d^\pi(s) \geq \epsilon^{k_0} \eta_{k_0}. \tag{15}$$

*Proof.* In vector form, we can write the stationary equation $d = dP_\pi$. Then, we also know that $d = dP_\pi^2$, and therefore $d \geq \epsilon^2 dP_u^2$ holds element-wise.

$$\begin{aligned}
d(s) &= \sum_{s',s''} d(s') P_\pi(s''|s') P_\pi(s|s''), \\
&\geq \sum_{s',s''} d(s') \epsilon^2 P_u(s''|s') P_u(s|s''), \\
&= \epsilon^2 \sum_{s'} d(s') P_u^2(s|s') \geq \epsilon^2 \eta_2.
\end{aligned}$$

One can also easily show that this property holds for any $k$-step, thus proving that $d(s) \geq \epsilon^k \eta_k$. $\square$

The previous result is important: for environments where $k_0$ is small, and the uniform policy is enough to guarantee high visitation rates (i.e., $\eta_{k_0}$ is not small), then all states are visited regularly.

### B.2.3. PROOF OF THEOREM 4.7 (RELAXED CHARACTERISTIC RATE)

*Proof of Theorem 4.7.* We prove the theorem considering the following general characteristic time

$$T_\epsilon(\omega; M)^{-1} := \inf_{\pi \in \Pi, r \in \mathcal{R}_\pi^\epsilon, M'_r \in \mathrm{Alt}_{\pi,r}^\epsilon(M)} \mathbb{E}_\omega[\mathrm{KL}_{P|P'_r}(s,a)],$$

and we also consider the single-policy case, since the extension to multi-policy follows immediately.

Start by noting that for all $r \in \mathcal{R}_\pi^\epsilon$ we have $\{M_r' : \|V_{M_r}^\pi - V_{M_r'}^\pi\|_\infty > 2\epsilon\} \subseteq \cup_s \{M_r' : |V_{M_r}^\pi(s) - V_{M_r'}^\pi(s)| > 2\epsilon\}$. Using this decomposition, we can show that

$$T_\epsilon(\omega; M)^{-1} \geq \inf_{r \in \mathcal{R}_\pi^\epsilon} \min_{s_0} \inf_{M_r' : |V_{M_r}^\pi(s_0) - V_{M_r'}^\pi(s_0)| > 2\epsilon} \mathbb{E}_{(s',a) \sim \omega}[\mathrm{KL}_{P|P_r'}(s',a)]$$

$$= \inf_{r \in \mathcal{R}_\pi^\epsilon} \min_{s_0} \inf_{M_r' : |V_{M_r}^\pi(s_0) - V_{M_r'}^\pi(s_0)| > 2\epsilon} \sum_{s',a} \omega(s',a) \mathrm{KL}_{P|P_r'}(s',a),$$

$$= \inf_{r \in \mathcal{R}_\pi^\epsilon} \min_{s_0} \inf_{M_r' : |V_{M_r}^\pi(s_0) - V_{M_r'}^\pi(s_0)| > 2\epsilon} \sum_s \omega(s,\pi(s)) \mathrm{KL}_{P|P_r'}(s,\pi(s)),$$

where we used the fact that the problem is unconstrained for state-action pairs $a \neq \pi(s)$. In fact, letting $\Delta V_r^\pi(s) = V_{M_r}^\pi(s) - V_{M_r'}^\pi(s)$ we have $\Delta V_r^\pi(s) = \gamma[\Delta P(s,\pi(s))^\top V_{M_r}^\pi + P_r'(s,\pi(s))^\top \Delta V_r^\pi]$, which only involves pairs of the type $(s,\pi(s))$. Alternatively, note that one can always claim

$$\inf_{M_r' : |V_{M_r}^\pi(s_0) - V_{M_r'}^\pi(s_0)| > 2\epsilon} \sum_{s',a} \omega(s',a) \mathrm{KL}_{P|P_r'}(s',a) \geq \inf_{M_r' : |V_{M_r}^\pi(s_0) - V_{M_r'}^\pi(s_0)| > 2\epsilon} \sum_s \omega(s,\pi(s)) \mathrm{KL}_{P|P_r'}(s,\pi(s)),$$

due to the non-negativity of the terms involved.

Then, observe the following inequality:

$$|\Delta V_r^\pi(s)| = \gamma \left| P(s,\pi(s))^\top V_{M_r}^\pi - P_r'(s,\pi(s))^\top V_{M_r'}^\pi \right|,$$

$$= \gamma \left| \Delta P(s,\pi(s))^\top V_{M_r}^\pi + P_r'(s,\pi(s))^\top \Delta V_r^\pi \right|,$$

$$\leq \gamma \left| \Delta P(s,\pi(s))^\top V_{M_r}^\pi \right| + \gamma \|\Delta V_r^\pi\|_\infty.$$

Since the inequality holds for all $s$, we derive $|\Delta V_r^\pi(s)| \leq \|\Delta V_r^\pi\|_\infty \leq \max_{s'} \frac{\gamma}{1-\gamma} \left| \Delta P(s',\pi(s'))^\top V_{M_r}^\pi \right|$, and thus

$$4\epsilon^2 < |\Delta V_r^\pi(s_0)|^2 \leq \|\Delta V_r^\pi\|_\infty^2 \leq \max_s \frac{\gamma^2}{(1-\gamma)^2} \left| \Delta P(s,\pi(s))^\top V_{M_r}^\pi \right|^2.$$

Now, note the following equality

$$\left| \Delta P(s,\pi(s))^\top V_{M_r}^\pi \right| = \left| \Delta P(s,\pi(s))^\top \rho_r^\pi(s) \right|,$$

Therefore, using Holder's inequality, that $\|p\|_1 \leq 2\|p\|_{TV}$ for a distribution $p$, and Pinsker's inequality, we obtain

$$4\epsilon^2 \leq \max_{s'} \frac{4\gamma^2}{(1-\gamma)^2} \|\Delta P(s',\pi(s'))\|_{TV}^2 \|\rho_r^\pi(s')\|_\infty^2,$$

$$\leq \max_{s'} \frac{2\gamma^2}{(1-\gamma)^2} \mathrm{KL}_{P|P_r'}(s',\pi(s')) \|\rho_r^\pi(s')\|_\infty^2,$$

$$= \max_{s'} \frac{2\gamma^2}{(1-\gamma)^2} \mathrm{KL}_{P|P_r'}(s',\pi(s')) \frac{\omega(s',\pi(s'))}{\omega(s',\pi(s'))} \|\rho_r^\pi(s')\|_\infty^2,$$

$$\leq \max_{s'} \frac{2\gamma^2}{(1-\gamma)^2} \mathrm{KL}_{P|P_r'}(s',\pi(s')) \omega(s',\pi(s')) \max_s \frac{\|\rho_r^\pi(s)\|_\infty^2}{\omega(s,\pi(s))}.$$

Therefore $\frac{2\epsilon^2(1-\gamma)^2}{\gamma^2} \min_s \frac{\omega(s,\pi(s))}{\|\rho_r^\pi(s)\|_\infty^2} \leq \max_{s'} \mathrm{KL}_{P|P_r'}(s',\pi(s')) \omega(s',\pi(s'))$, and thus

$$\sum_s \omega(s,\pi(s)) \mathrm{KL}_{P|P_r'}(s,\pi(s)) \geq \max_{s'} \mathrm{KL}_{P|P_r'}(s',\pi(s')) \omega(s',\pi(s')) \geq \frac{2\epsilon^2(1-\gamma)^2}{\gamma^2} \min_s \frac{\omega(s,\pi(s))}{\|\rho_r^\pi(s)\|_\infty^2}.$$

Using this inequality in the initial lower bound of $T_\epsilon(\omega; M)^{-1}$ we find

$$T_\epsilon(\omega; M)^{-1} \geq \inf_{r \in \mathcal{R}_\pi^\epsilon} \min_s 2\epsilon^2 \frac{(1-\gamma)^2 \omega(s,\pi(s))}{\gamma^2 \|\rho_r^\pi(s)\|_\infty^2},$$

which concludes the proof. $\qquad \square$

### B.2.4. PROOF OF COROLLARY 4.8

*Proof of Corollary 4.8.* We begin by rewriting the following optimization problem of $U_\epsilon(\omega; M)$. Note that

$$\sup_{r \in \mathcal{R}_\pi} \max_s \|\rho_r^\pi(s)\|_\infty^2 = \max_s \sup_{r \in \mathcal{R}_\pi} \max_{s'} |\rho_r^\pi(s, s')|^2,$$

$$= \max_s \max_{s'} \left( \sup_{r \in \mathcal{R}_\pi} |\rho_r^\pi(s, s')| \right)^2,$$

$$= \max_s \max_{s'} \max \left( \sup_{r \in \mathcal{R}_\pi} \rho_r^\pi(s, s'), \sup_{r \in \mathcal{R}_\pi} -\rho_r^\pi(s, s') \right)^2.$$

From the proof of Lemma B.6 we can derive the following expression $\rho_r^\pi(s, s') = e_{s'}^\top \Gamma^\pi(s) r$, where

$$\Gamma^\pi(s) := K^\pi(s) G^\pi, \quad K^\pi(s) := (I - \mathbf{1} P(s, \pi(s))^\top), \quad G^\pi := (I - \gamma P^\pi)^{-1},$$

and $e_{s'}$ is the $s'$-th element of the canonical basis in $\mathbb{R}^S$. Hence, the optimization problem $\sup_{r \in \mathcal{R}} \rho_r^\pi(s, s')$ is a linear program.

In the last part of the proof we consider the case where $\mathcal{R}_\pi = [0, 1]^S$. We exploit the following fact that holds for any vector $y \in \mathbb{R}^n$: $\max_{x \in [0,1]^n} y^\top x = \sum_{i:y_i>0} y_i$. Then, let $\Gamma_{ij}^\pi(s) := (K^\pi(s) G^\pi)_{ij}$ and define

$$\Gamma_+^\pi(s, s') := \sum_{j:\Gamma_{s',j}^\pi(s)>0} \Gamma_{s',j}^\pi(s),$$

and, similarly, $\Gamma_-^\pi(s, s') := -\sum_{j:\Gamma_{s',j}^\pi(s)<0} \Gamma_{s',j}^\pi(s)$. Hence, we have

$$\max_{r \in [0,1]^S} |\rho_r^\pi(s, s')| = \max \left( \Gamma_+^\pi(s, s'), \Gamma_-^\pi(s, s') \right).$$

$\square$

### B.2.5. COMPUTING THE OPTIMAL RELAXED CHARACTERISTIC RATE $U^\star$

For a general polytope of rewards, defined by $(A, b)$, one can write the following program to solve $v_\pi(s, s') = \sup_{r \in \mathcal{R}} \rho_r^\pi(s, s')$:

$$v_\pi(s, s') := \max_{r \in [0,1]^S, t \in \mathbb{R}} \quad t$$
$$\text{subject to} \quad e_{s'}^\top \Gamma^\pi(s) r \geq t, \tag{16}$$
$$A r \leq b.$$

This formulation is a linear program since the objective and all constraints are linear in the decision variables. The polytope can represent various convex sets, including hypercubes, simplices, or other polyhedral shapes, depending on the specific application. In contrast, the optimization for the reward-free case is straightforward, which follows from the result in Corollary 4.8.

Therefore, given a collection of deterministic target policies $\{\pi_i\}_{i=1}^N$ and reward sets $\{\mathcal{R}_i\}_{i=1}^N$ (finite, or a convex set), we precompute, for every policy $i$ and state $s$,

$$A_i(s) = \max_{s'} \max \left( \sup_{r \in \mathcal{R}_i} e_{s'}^\top \Gamma^{\pi_i}(s) r, \sup_{r \in \mathcal{R}_i} -e_{s'}^\top \Gamma^{\pi_i}(s) r \right)^2,$$

The inner sup is solved twice in the general case. For a finite reward set it is a direct maximum, and one can directly compute $A_i(s) = \max_{s'} \max_{r \in \mathcal{R}_i} |\rho_r^\pi(s, s')|^2$. For the reward free case we can use the result from Corollary 4.8 to obtain $A_i(s) = \max_{s'} \max \left( \Gamma_+^{\pi_i}(s, s'), \Gamma_-^{\pi_i}(s, s') \right)^2$.

Stacking the rows $A_i$ yields a matrix $A \in \mathbb{R}^{N \times |\mathcal{S}|}$. For a stationary occupancy measure $\omega \in \Omega(M)$ we define $f_i(\omega) = \max_{s \in \mathcal{S}} A_i(s) \dfrac{\gamma^2}{2\epsilon^2(1-\gamma)^2} \dfrac{1}{\omega(s, \pi_i(s))}$. The optimal relaxed rate is obtained from the convex optimisation problem

$$U_\epsilon^\star = \min_{\omega \in \Delta(\mathcal{S} \times \mathcal{A})} \max_{i=1,\ldots,N} f_i(\omega) \quad \text{s.t.} \sum_a \omega(s, a) = \sum_{s',a'} \omega(s', a') P(s \mid s', a') \ \forall s.$$

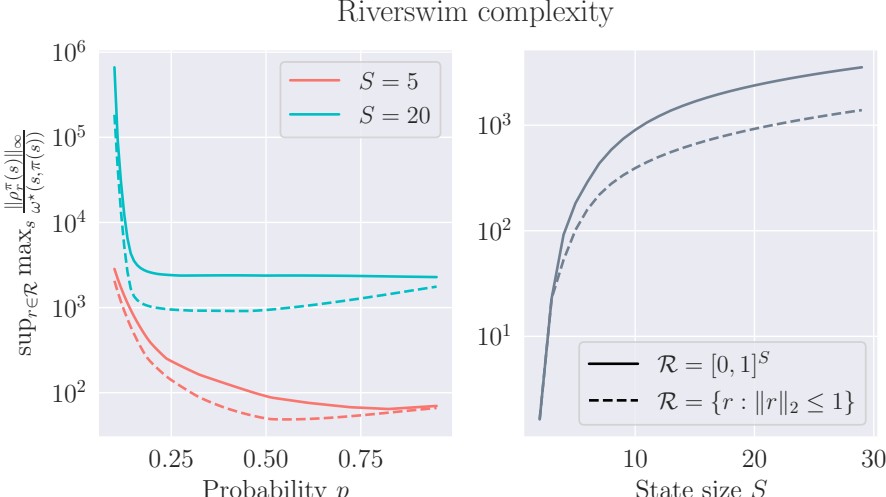

*Figure 5.* Complexity of Riverswim for varying reward sets and state space sizes for a single policy $\pi(\cdot) = a_2$ that moves towards the river's end. Left: complexity for different $p$ values (prob. of moving toward the river's end). Right: complexity for $p = 0.3$ with varying state space sizes. Solid curves evaluate on $\mathcal{R} = [0, 1]^S$, dashed curves on $\mathcal{R} = \{r \in [0, 1]^S : \|r\|_2 \leq 1\}$.

We solve this minimax programme with CVXPY, initialising $\omega$ from the previous run to warm-start the solver (Diamond & Boyd, 2016).

**Scaling of Reward-free Sample Complexity - An Example**   As an example, in Figure 5 we show the policy evaluation sample complexity in the Riverswim environment (Strehl & Littman, 2004) for a single policy $\pi$. In this environment the agent swims towards the river's end, while the river's current opposes the movement of the agent. The policy tries to move the agent toward the river's end, and, as $p$ (the probability of moving toward the river's end) decreases, the reward-free sample complexity increases.

# C. `MR-NaS` and Numerical Results

## C.1. MR-NaS Proof of Theorem 5.2

*Proof of Theorem 5.2.* Let $\mathcal{C}_{\pi,r} = \{\|\hat{V}_r^\pi - V_r^\pi\|_\infty > \epsilon\}$ be the event that at the stopping time the value estimate is not $\epsilon$-accurate in $\pi, r \in \mathcal{R}_\pi$. Hence, we have that $M \in \mathrm{Alt}_{\pi,r}^{\epsilon/2}(M_\tau)$.

Therefore, we can say that

$$\mathcal{C} = \{\exists \pi \in \Pi, r \in \mathcal{R}_\pi : \|\hat{V}_r^\pi - V_r^\pi\|_\infty > \epsilon\} \subset \{\exists \pi \in \Pi, r \in \mathcal{R}_\pi : M \in \mathrm{Alt}_{\pi,r}^{\epsilon/2}(M_\tau)\} := \mathcal{B}.$$

Then, we obtain the following chain of inequalities:

$$
\begin{aligned}
\mathbb{P}_M\left(\tau_\delta < \infty, \mathcal{C}\right) &\leq \mathbb{P}_M\left(\exists t \geq 1 : tU_{\epsilon/2}(N_t/t; M_t)^{-1} \geq \beta(N_t, \delta), \mathcal{B}\right), \\
&\leq \mathbb{P}_M\left(\exists t \geq 1 : tT_{\epsilon/2}(N_t/t; M_t)^{-1} \geq \beta(N_t, \delta), \mathcal{B}\right), \\
&= \mathbb{P}_M\left(\exists t \geq 1 : \inf_{\pi,r \in \mathcal{R}_\pi, M' \in \mathrm{Alt}_{\pi,r}^{\epsilon/2}(M_t)} \sum_{s,a} N_t(s,a)\mathrm{KL}_{M_t|M'}(s,a) \geq \beta(N_t, \delta), \mathcal{B}\right), \\
&\leq \mathbb{P}_M\left(\exists t \geq 1 : \sum_{s,a} N_t(s,a)\mathrm{KL}_{M_t|M}(s,a) \geq \beta(N_t, \delta)\right), \\
&\leq \delta,
\end{aligned}
$$

where the conclusion follows from Jonsson et al. (2020, Prop. 1).

For the sample complexity, we use the following facts:

1. The forcing policy is chosen as $\pi_{f,t}(\cdot|s) = \mathrm{softmax}\left(-\beta_t(s)N_t(s,\cdot)\right)$ with $\beta_t(s) = \frac{\beta \log(N_t(s))}{\max(1, \max_a |N_t(s,a) - \min_b N_t(s,b)|)}, \beta \in [0,1]$ and $(\mathrm{softmax}(x))_i = e^{x_i}/\sum_j e^{x_j}$ for a vector $x$. This choice encourages to select under-sampled actions for $\beta > 0$, while for $\beta = 0$ we obtain a uniform forcing policy $\pi_{f,t}(a|s) = 1/A$. We then mix $\omega_t^\star$ with $\pi_{f,t}$ using a mixing factor $\epsilon_t = 1/\max(1, N_t(s_t))^\alpha$, with $N_t(s) = \sum_a N_t(s,a)$. The values $\alpha, \beta$ need to guarantee $\alpha + \beta \leq 1$ (Russo & Vannella, 2024). Under this choice the chain induced by $\pi_t$ is ergodic.

2. We also note that the optimal solution $\omega^\star$ satisfies $\omega^\star(s,a) > 0$ for all $(s,a)$. Then, such solution induces an ergodic (irreducible and aperiodic) chain by Assumption 3.1 and Assumption 5.1.

3. We note that the solution $\omega^\star$ is unique by Assumption 5.1.

4. Then, the sample complexity results follow from noting that $U_{\epsilon/2}(\omega; M) = 4U_\epsilon(\omega; M)$ and applying the same methods as in Russo & Vannella (2024, Theorem 3.3) mutatis mutandis (which follows the proof technique of (Al Marjani et al., 2021)).

$\square$

## C.2. Environment Details

In this section we delve more into the detail of the numerical results for the tabular case. We focus on different hard-exploration tabular environments: `Riverswim` (Strehl & Littman, 2004), `Forked Riverswim` (Russo & Proutiere, 2023a), `DoubleChain` (Kaufmann et al., 2021) and `NArms` (Strehl & Littman, 2004) (an adaptation of `SixArms` to $N$ arms). Here we provide a brief description of the environments.

**Riverswim.** The RiverSwim environment is a classic reinforcement learning benchmark designed to test exploration (Strehl & Littman, 2004). It consists of a series of states arranged in a linear chain, where an agent can choose to swim right (downstream) or left (upstream). In the single-reward setting the agent can achieve a positive return by swimming right, but requires overcoming a strong current, making it a less probable event. Conversely, swimming left generally offers small to zero rewards, but is easier. This setup requires the agent to balance immediate, safer rewards with potentially higher but riskier returns. It is exponentially hard for a random uniform agent to reach the final state.

In figure Figure 6 is shown a depiction of the environment. There are $n$ states, and two main parameters, $p, p' \in (0,1)$, and their sum $p_{\mathrm{sum}} = p + p' < 1$. In the figure, each tuple $(a, p)$ represents the action $a$ that triggers the transition and the

probability $p$ of that event. The agent starts in state $s_0$, and in every state can only take two actions $\{a_0, a_1\}$. For small values of $p$ it becomes difficult for the agent to swim right (i.e., reach $s_{n-1}$), and larger values of $p'$ can also hinder the progress. On the other hand, swimming towards the left is easier, since the probability of $P(s_{i-1}|s_i, a_0) = 1$. For the experiments, we used $n \in \{15, 20, 30\}, p = 0.7, p' = 6(1-p)/7$ .

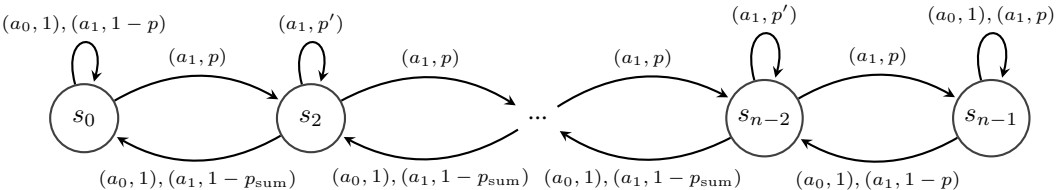

*Figure 6.* `Riverswim` environment (Strehl & Littman, 2004). Each tuple $(a, p)$ represents the action $a$ that triggers the transition and the probability $p$ of that event.

**Forked Riverswim.** The Forked RiverSwim environment (Russo & Proutiere, 2023a) is a variation of the traditional RiverSwim reinforcement learning benchmark, designed to test more complex exploration strategies. In this variant, the state space branches into multiple paths, resembling a river that forks. At intermediate states the agent can switch between the forks, while the end states are not connected. This variant requires the agent to make more sophisticated decisions to explore the environment. This setup increases the sample complexity and challenges the agent's ability to generalize across different paths within the environment.

In Figure 7 is shown a depiction of the environment. There are a total of $2n + 1$ states, and two parameters $p, p' \in (0, 1)$, so that $p_{\text{sum}} = p + p' < 1$. In the figure, each tuple $(a, p)$ represents the action $a$ that triggers the transition and the probability $p$ of that event. The agent starts in state $s_0$, and in every state can chose between three actions $\{a_0, a_1, a_2\}$. For small values of $p$ it becomes difficult for the agent to swim right in both forks, and larger values of $p'$ can also hinder the progress. As in Riverswim, swimming towards the left is easier, since the probability of $P(s_{i-1}|s_i, a_0) = 1$. For the experiments, we used $n \in \{8, 10, 15\}, p = 0.7, p' = 6(1-p)/7$.

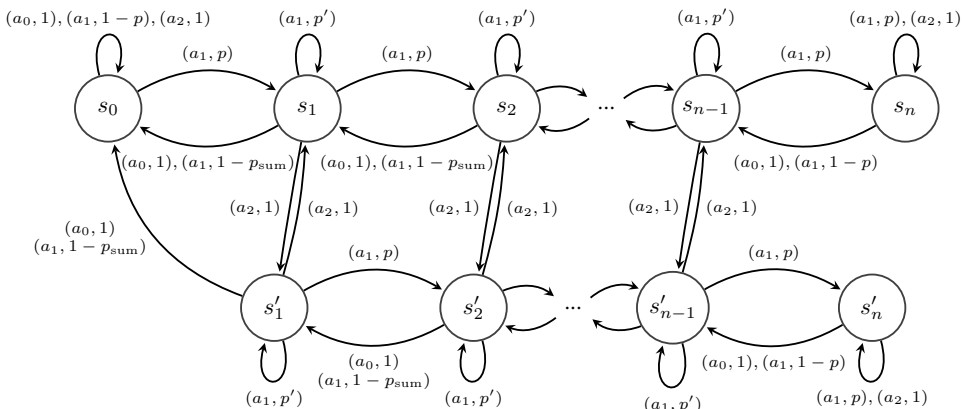

*Figure 7.* `Forked Riverswim` environment (Russo & Proutiere, 2023a). Each tuple $(a, p)$ represents the action $a$ that triggers the transition and the probability $p$ of that event.

**Double Chain.** The Double Chain environment (Kaufmann et al., 2021) consists of two chains, similarly to the Forked Riverswim. The main difference consists in the fact that it is not possible to switch between the two chains, and intermediate states are transient (there is no parameter $p'$).

In Figure 8 is shown a depiction of the environment. There are a total of $2n + 1$ states, and one parameters $p \in (0, 1)$. In the figure, each tuple $(a, p)$ represents the action $a$ that triggers the transition and the probability $p$ of that event. The agent starts in state $s_0$, and in every state can chose between two actions $\{a_0, a_1\}$. For small values of $p$ it becomes difficult for the agent to move to the end of the chain in both chains. For the experiments, we used $n \in \{8, 10, 15\}, p = 0.7$.

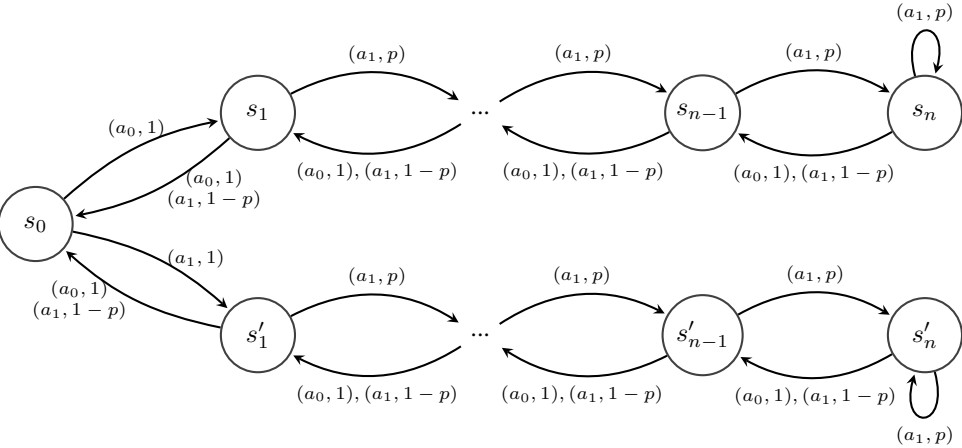

*Figure 8.* `Double Chain` environment ([Kaufmann et al., 2021]) . Each tuple $(a, p)$ represents the action $a$ that triggers the transition and the probability $p$ of that event.

**NArms.** This environment is an adaptation to $N$ arms of the original `6Arms` environment from ([Strehl & Littman, 2004]). Differently from the previous environments, this is a bandit-like environment, where the agent is presented with $n$ different actions (or arms) to choose from. The agent starts in a state $s_0$ and selects an arm $a_i$. Upon selecting an arm, the agent may transition to corresponding state $s_i$. Certain arms are more difficult to observe, in the sense that the transition probability is lower. This property mimics the probability of collecting a reward in a bandit problem. In [Figure 9] is shown a depiction of the environment. There are a total of $n + 1$ states, and one parameters $p_0 \in (0, 1)$. In the figure, each tuple $(a, p)$ represents the action $a$ that triggers the transition and the probability $p$ of that event. The notation $a_{n_0:n_0+n}$ indicates all the actions in $\{a_{n_0}, \ldots, a_{n_0+n}\}$. The agent starts in state $s_0$, and in every state she can select between $n$ actions $\{a_0, a_1, \ldots, a_{n-1}\}$. For small values of $p_0$ it becomes difficult for the agent to move to different states. Similarly, it is harder to navigate to states $s_i$ for large values of $n$. We used $p_0 = 0.7$ and $n \in \{10, 20, 30\}$.

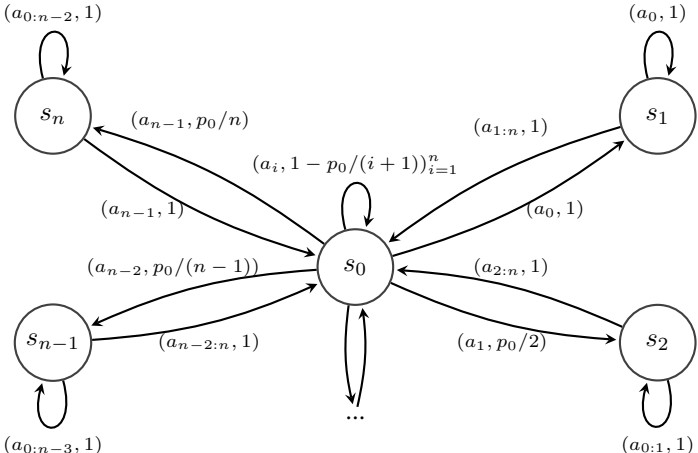

*Figure 9.* `NArms` environment. Each tuple $(a, p)$ represents the action $a$ that triggers the transition and the probability $p$ of that event. In the figure the notation $a_{n_0:n_0+n}$ indicates all the actions in $\{a_{n_0}, \ldots, a_{n_0+n}\}$. In state $s_0$ the probability to remain in $s_0$ for any action $a_i$ is $P(s_0|s_0, a_i) = 1 - p_0/(i + 1)$, with the exception that $P(s_0|s_0, a_0) = 0$.

## C.3. Algorithm Details

In this section we briefly explain the mechanisms of each algorithm used in the experiments, and, in case, their adaptation. Note that for all algorithms we evaluated the policies by using the MDP estimate $M_t$ through policy evaluation.

**Noisy Policy (Uniform).** This method simply computes a mixture of the target policies $\pi_{\text{mix}}(a|s) = \frac{|\{\pi \in \Pi : \pi(s) = a\}|}{|\Pi|}$, which is then mixed with a uniform policy $\pi_u$ with a constant mixing factor $\varepsilon_t = 0.3$. The resulting behavior policy is $\pi_b = (1 - \varepsilon_t)\pi_{\text{mix}} + \varepsilon_t \pi_u$.

**Noisy Policy (Visitation).** This method is similar to the previous one, but the mixing factor is not constant anymore. We take a mixing factor that is $\epsilon_t = 1/N_t(s_t)$, which based on the number of visits to state $s_t$. The resulting behavior policy is $\pi_b = (1 - \varepsilon_t)\pi_{\text{mix}} + \varepsilon_t \pi_u$.

**SF-NR (McLeod et al., 2021).** This is an algorithm for multi-task policy evaluation based on the Successor Representation. The pseudo-steps of the algorithm can be found in Algorithm 2. The method maintains a successor representation $\psi_{\pi,t}$ for each policy $\pi$, as well as a behavior policy $\pi_\beta$. These are learned using TD-learning, and the behavior policy uses the variation between $\psi_{\beta,t+1}$ and $\psi_{\beta,t}$ as a reward. In our experiment we used a temperature $T = 2$ and a discount factor for the successor representation $\gamma_\psi = 0.99$.

---

**Algorithm 2** `SF-NR`

---

**Require:** Discount factor $\gamma$; Temperature $T$; Successor discount factor $\gamma_\psi$; policy set $\Pi$.
1: Set $\pi_\beta(\cdot|s) = \mathcal{U}(\{1, \ldots, A\})$ for all states $s$.
2: Set $\psi_{\pi,1}(s,a) = 1$ for all $(s,a)$.
3: **while** not done **do**
4:     Compute $\hat{\pi}_\beta(\cdot|s_t) = \text{Softmax}(\pi_\beta(\cdot|s_t)/T)$
5:     Sample $a_t$ from $(1 - \varepsilon_t)\hat{\pi}_\beta(\cdot|s_t) + \varepsilon_t/A$ and observe $s_{t+1} \sim P(\cdot|s_t, a_t)$.
6:     **for** $\pi \in \Pi$ **do**
7:         Compute $\delta_t = 1 + \gamma_\psi \psi_{\pi,t}(s_{t+1}, \pi(s_{t+1})) - \psi_{\pi,t}(s_t, a_t)$
8:         Set $\psi_{\pi,t+1}(s_t, a_t) = \psi_{\pi,t}(s_t, a_t) + \alpha_t \delta_t$, where $\alpha_t = 1/N_t(s_t, a_t)$.
9:     **end for**
10:    Compute $\delta_{\psi,t} = 1/|\Pi| \sum_{\pi \in \Pi} \|\text{Vec}(\psi_{\pi,t+1}) - \text{Vec}(\psi_{\pi,t})\|_1$.
11:    Update $\pi_\beta(a_t|s_t) \leftarrow \pi_\beta(a_t|s_t) + \alpha_t (\delta_{\psi,t} + \gamma \max_a \pi_\beta(a|s_{t+1}) - \pi_\beta(a_t|s_t))$, where $\alpha_t = 1/N_t(s_t, a_t)$.
12:    Update MDP estimate $M_t$ and set $t \leftarrow t + 1$.
13: **end while**

---

**GVFExplorer (Jain et al., 2024).** This method considers variance-based exploration strategy for learning general value functions (Sutton et al., 2011) based on minimizing the MSE. The pseudo-steps of the algorithm can be found in Algorithm 3. Given the current estimate of the MDP $M_t$, the method estimates $\text{Var}_{s,a}^\pi(t)$, the variance of the return $G^\pi = r_1 + \gamma r_2 + \gamma^2 r_3 + \ldots$ under $\pi$ staring from $(s,a)$, $\forall \pi \in \Pi, (s,a) \in \mathcal{S} \times \mathcal{A}$. Then, a behavior policy is computed as $\pi_\beta(a|s) = \frac{\sqrt{\sum_{\pi \in \Pi} \pi(a|s)\text{Var}_{s,a}^\pi(t)}}{\sum_b \sqrt{\sum_{\pi \in \Pi} \pi(b|s)\text{Var}_{s,b}^\pi(t)}}$. Lastly, we mix this policy with a uniform policy. For this method we used a fixed mixing factor $\varepsilon = 0.3$ (we did not use a visitation based mixing factor because performance deteriorated).

---

**Algorithm 3** `GVFExplorer`

---

**Require:** Mixing factor $\varepsilon$, policy set $\Pi$.
1: Set $\text{Var}_{s,a}^\pi(1) = 1$ for all $(s,a) \in \mathcal{S} \times \mathcal{A}, \pi \in \Pi$.
2: **while** not done **do**
3:     Set $\pi_\beta(a|s) = \frac{\sqrt{\sum_{\pi \in \Pi} \pi(a|s)\text{Var}_{s,a}^\pi(t)}}{\sum_b \sqrt{\sum_{\pi \in \Pi} \pi(b|s)\text{Var}_{s,b}^\pi(t)}}$
4:     Sample $a_t$ from $(1 - \varepsilon)\pi_\beta(\cdot|s_t) + \varepsilon/A$ and observe $s_{t+1} \sim P(\cdot|s_t, a_t)$.
5:     Update MDP estimate $M_t$, variance estimates $\{\text{Var}^\pi(t)\}_{\pi \in \Pi}$ and set $t \leftarrow t + 1$.
6: **end while**

---

Note that $\pi_\beta$ is similar to the generative solution in Equation (2) (i.e., $\Omega(M) = \Delta(\mathcal{S} \times \mathcal{A})$). In fact, `GVFExplorer` neglects the forward equations when deriving $\pi_\beta$. The resulting solution does not take into account the dynamics induced by the behavior policy, effectively making it a generative method (compare with the generative solution proved in (Al Marjani & Proutiere, 2021; Russo & Proutiere, 2023a)). We believe this is due to a term being omitted in the proof of Jain et al. (2024, Theorem 4.1) that accounts for how changes in the state distribution induced by the behavior policy impact the variance.

**MR-NaS.** For `MR-NaS` we computed the exploration strategy $\omega_t^\star$ every 500 steps in the environment, to avoid excessive computational burden. The policy is then mixed with a forcing policy that is $\pi_{f,t}(\cdot|s) = \text{softmax}(-\beta_t(s)N_t(s,\cdot))$ with $\beta_t(s) = \frac{\beta \log(N_t(s))}{\max_a |N_t(s,a) - \min_b N_t(s,b)|}, \beta \in [0,1]$ and $(\text{softmax}(x))_i = e^{x_i}/\sum_j e^{x_j}$ for a vector $x$. This choice encourages to

select under-sampled actions for $\beta > 0$, while for $\beta = 0$ we obtain a uniform forcing policy $\pi_{f,t}(a|s) = 1/A$. We then mix $\omega_t^\star$ with $\pi_{f,t}$ using a mixing factor $\epsilon_t = 1/\max(1, N_t(s_t))^\alpha$, with $N_t(s) = \sum_a N_t(s,a)$. The values $\alpha, \beta$ need to guarantee $\alpha + \beta \leq 1$ (Russo & Vannella, 2024), hence we chose $\alpha = 0.99$ and $\beta = 0.01$.

### C.4. Additional Results and Experimental Details

In this sub-section we report additional results. To run reproduce the results, we refer the reader to the README.md file in the supplementary material. In Figure 10 are reported the results for the multi-reward multi-policy case with various sizes of the state space. Similarly, in Figure 11 are reported the reward-free results for the multi-policy case, and in Figure 12 the results for the single-policy reward-free case. Experiments were run over $10^6$ time-steps, with 30 seeds. Confidence intervals were computed using bootstrap (Efron, 1992).

**Experimental Details: target policies.** In all experiments we fix a discount factor $\gamma$ and run each simulation for $T$ time steps. In the multi-policy scenario, for each random seed, we choose $m = 3$ target policies as follows:

1. Draw without replacement three state-action pairs $(s_i^*, a_i^*)_{i=1}^3$ uniformly from $\mathcal{S} \times \mathcal{A}$.

2. Define the one-hot reward $r_i(s,a) = \mathbf{1}\{(s,a) = (s_i^*, a_i^*)\}$ and compute the target policy $\pi_i$ is computed via policy iteration on $(M, r_i)$.

On the other hand, in the single-policy scenario we use a default target policy policy $\pi_{\text{def}}$ that is different for each environment. Concretely, we solve policy iteration on a fixed one-hot reward $r_{\text{def}}$ where $r_{\text{def}}(s,a) = \mathbf{1}_{\{(s,a)=(s^\star, a^\star)\}}$ for some state $(s^\star, a^\star)$. Then, the target policy is computed via policy iteration on $(M, r_{\text{def}})$.

In particular, for Riverswim we have $(s^*, a^*) = (s_{n-1}, a_1)$; for Forked Riverswim we have $(s^*, a^*) = (s_n', a_1)$; for Double chain we have $(s^*, a^*) = (s_n', a_1)$; for NArms we have $(s^*, a^*) = (s_n, a_{n-1})$. See also Appendix C.2 for more details on the environments.

**Experimental Details: reward sets.** We use the following reward sets: finite rewards and the reward-free scenario.

In the **finite reward-set** scenario, we restrict $\mathcal{R} = \{r_i\}_i$ to a uniformly chosen subset of size 3 from the canonical basis $\mathcal{R}_{\text{canon}}$ (sampled without replacement).

In the **reward-free** scenario, we use **MR-NaS** with the result in Corollary 4.8 and evaluate the collected data over the entire basis $\mathcal{R}_{\text{canon}}$ for each $\pi_i$.

**Experimental Details: evaluation protocol.** During each run of $T$ steps, the agent collects transitions and updates its empirical model $\hat{P}_t$. Every $F$ steps, we perform a batched evaluation:

1. For each reward set (finite or reward-free) and each policy $\pi$, compute the true value $V_{M_r}^\pi$ offline via value iteration on the known dynamics. For the reward free case we use $\mathcal{R}_{\text{canon}}$ to perform evaluation.

2. Compute the estimated value $\hat{V}_r^\pi(t)$ by performing policy evaluation on $\hat{P}_t$ over the reward sets.

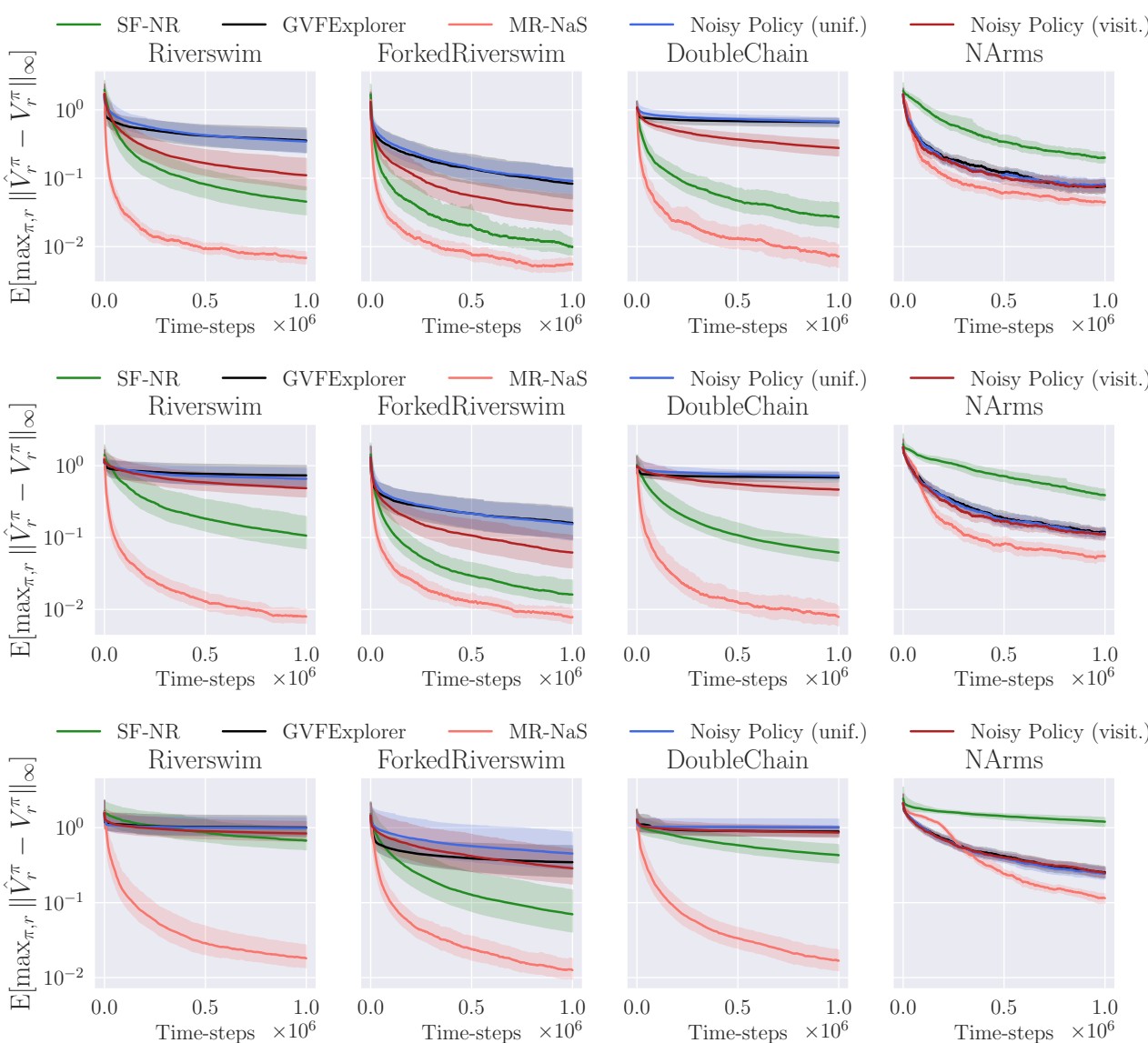

*Figure 10.* Multi-reward multi-policy evaluation for different sizes of the MDPs: from top to bottom the state space size is $15, 20, 30$. Shaded curves represent 95% confidence intervals.

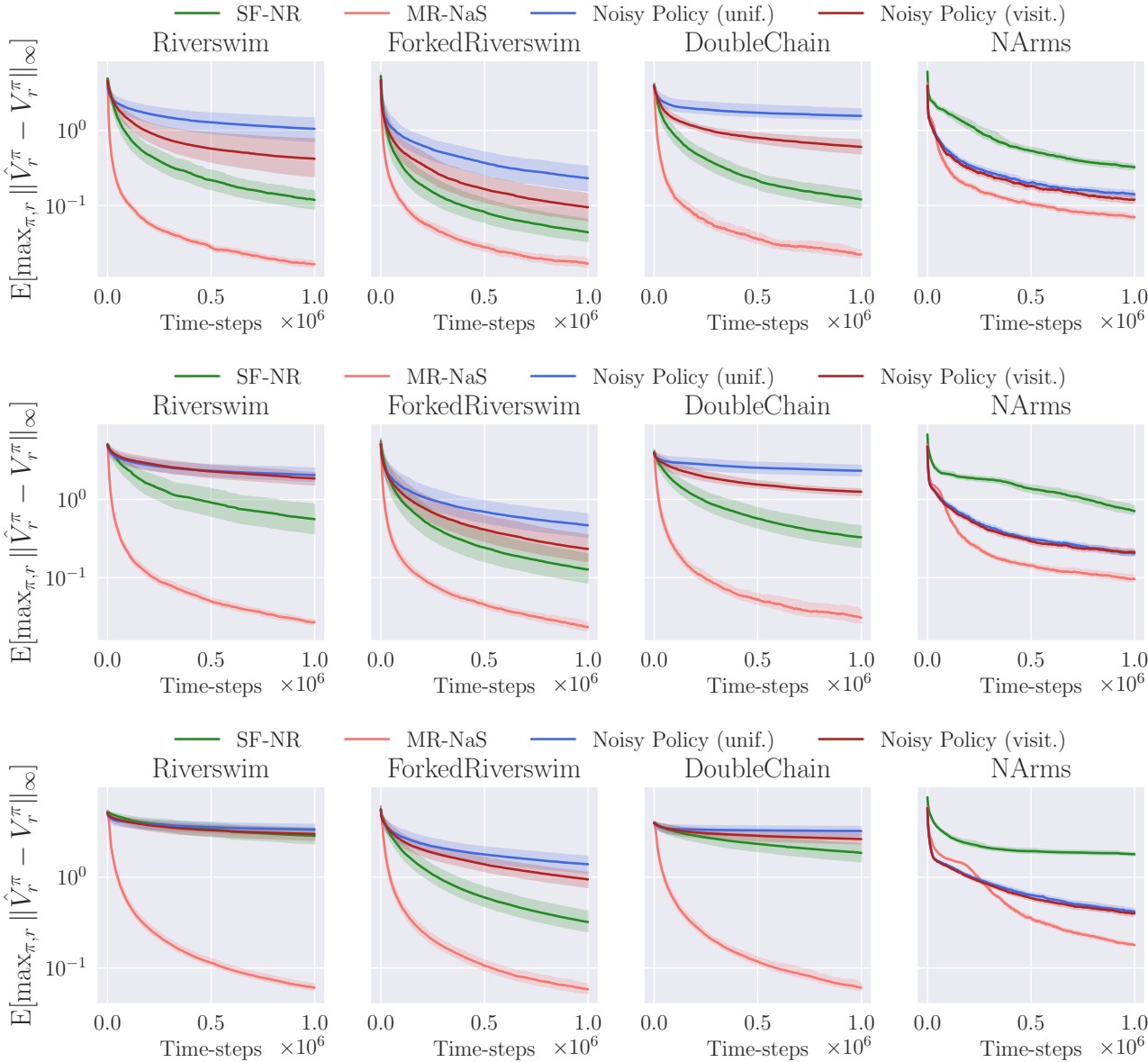

*Figure 11.* Reward-Free multi-policy evaluation for different sizes of the MDPs: from top to bottom the state space size is 15, 20, 30. Here we depict the average error over the canonical basis $\mathcal{R}_{\text{canonical}}$. Shaded curves represent 95% confidence intervals.

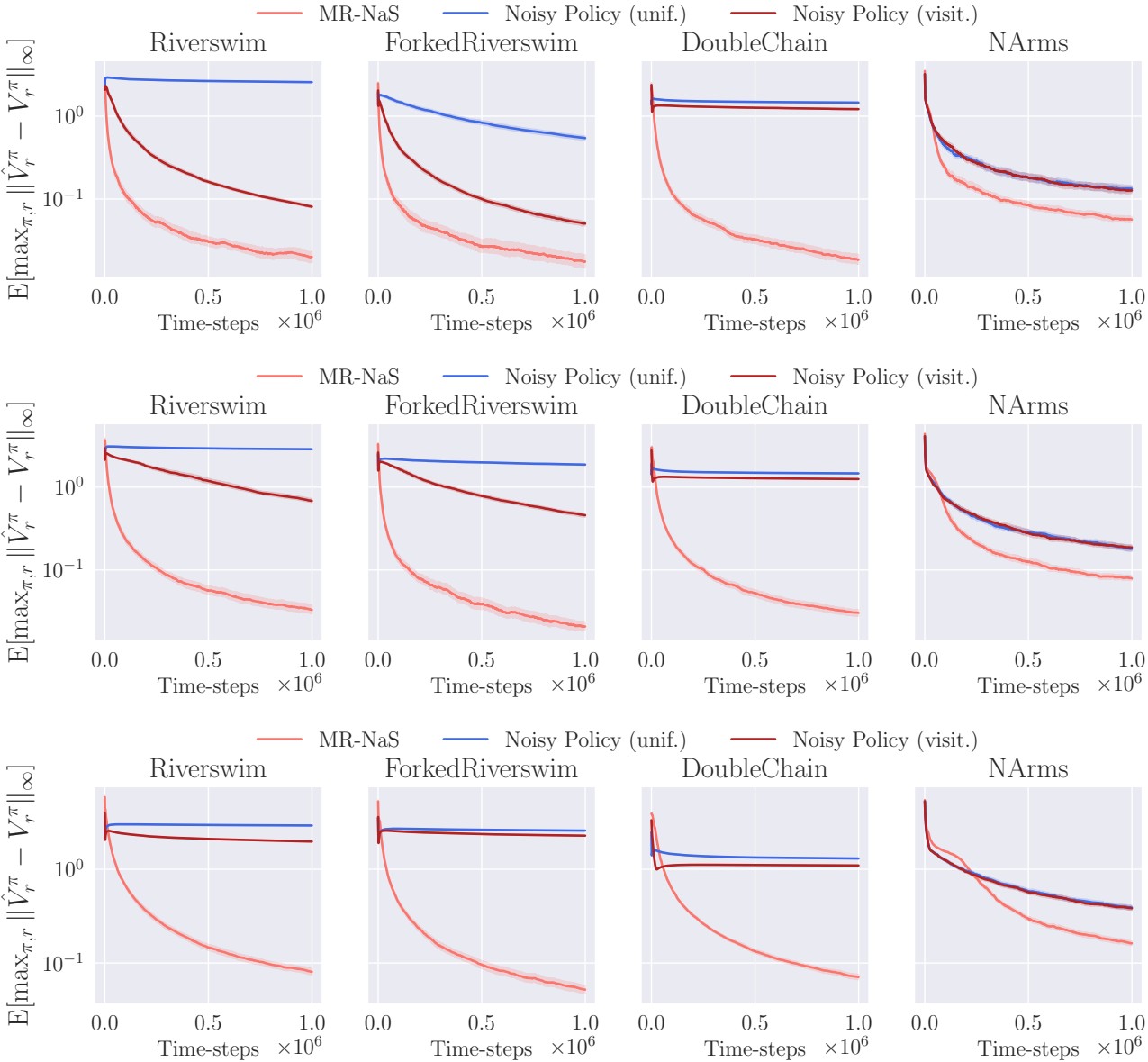

*Figure 12.* Reward-Free single-policy evaluation for different sizes of the MDPs: from top to bottom the state space size is $15, 20, 30$. Here we depict the average error over the canonical basis $\mathcal{R}_{\text{canonical}}$. Shaded curves represent 95% confidence intervals.

