# OpenReview forum: "Adaptive Exploration for Multi-Reward Multi-Policy Evaluation"
_ICML.cc/2025/Conference — ICML 2025 poster_

### Official Review · Reviewer_pv4R · 2025-03-10

**Overall Recommendation:** 3

**Summary:**

This paper studies sample-efficient exploration for the multi-reward multi-policy evaluation problem, which aims to simultaneously evaluate all target policy-reward pairs. The authors use an instance-specific lower bound to guide the design of their proposed efficient exploration method. Furthermore, they propose a convex approximation for bound computation and present numerical simulations to demonstrate the effectiveness of their method.

**Claims And Evidence:**

It seems that the claims are supported by sufficient evidence and interpretation.

**Essential References Not Discussed:**

There are no essential related works that are missing from the citations or discussion in the paper.

**Experimental Designs Or Analyses:**

The experimental designs and analyses seem sound and valid.

**Methods And Evaluation Criteria:**

The proposed methods and evaluation criteria seem reasonable for the problem and application at hand.

**Other Comments Or Suggestions:**

1. Could this be a potential typo?:
In the line 176 second column, Alt=\empty is empty -> Alt is empty?

2. Could this be a potential typo?:
In the line 2 in Algorithm 1, inf -> arginf?

**Other Strengths And Weaknesses:**

Strength

    1. This paper provides an instance-specific sample complexity lower bound that separately considers P and r. Furthermore, based on this lower bound, the authors identify the main factor that influences the scaling of the sample complexity lower bound.

    2. To address the non-convexity of the lower bound, the authors propose an appropriate convex envelope.

Weakness

    1. It would be helpful if the difference or relationship between this work and the main prior work, MR-NaS, were explained more clearly.

**Questions For Authors:**

1. In my understanding, the first bullet below Proposition 4.1 suggests that if the condition is satisfied, then Alt is empty, leading to a no-confusing scenario. Is my understanding correct, or am I missing something?
    2. I would appreciate a clarification on how the authors' algorithm differs from the prior work MR-NaS in terms of algorithmic aspects, as I had difficulty understanding this distinction.
    3. If there are no algorithmic differences, should I understand the contribution as providing a theoretical sample complexity bound for applying MR-NaS to the Multi-Reward Multi-Policy Evaluation problem?

**Relation To Broader Scientific Literature:**

Efficient multi-policy evaluation is useful for conducting ablations or tuning various algorithm hyperparameters, while efficient multi-reward evaluation can serve as a subroutine in MORL algorithms.

**Theoretical Claims:**

I did not check the proofs in detail but reviewed the insights behind the claims.

---

> ### Author Rebuttal · Authors · 2025-03-28
>
> We would like to thanks the reviewer evaluating our work and their positive appreciation of our paper. Below, we address your  questions:
>
> > In my understanding, the first bullet below Proposition 4.1 suggests that if the condition is satisfied, then Alt is empty, leading to a no-confusing scenario. Is my understanding correct, or am I missing something?
>
>  The reviewer's understanding is correct: as $\epsilon$ increases or $\gamma$ decreases, the condition is more likely to hold, which leads to a no-confusing scenario. In the text, we mistakenly wrote the opposite due to a last-minute change in the direction of the inequality in Proposition 4.1. We will correct this in the final version of the paper.
>
> > I would appreciate a clarification on how the authors' algorithm differs from the prior work MR-NaS [...] If there are no algorithmic differences, should I understand the contribution as providing a theoretical sample complexity bound for applying MR-NaS to the Multi-Reward Multi-Policy Evaluation problem?
>
>
> We thank the reviewer for the opportunity to clarify this point.
>
> This algorithmic structure is actually inspired by approaches in Best Arm Identification in Bandit problems (Garivier \& Kaufmann, 2016), where it is common to apply a Track-and-Stop procedure. This procedure computes   an exploration strategy $w_t^*$ at each time step and sample the next action from it (for more examples, see also Degenne \& Koolen, 2019; Al Marjani et al., 2021).
>
>
> While the general algorithmic structure is similar in all these works, the specific optimization problem in computing $w_t^*$ differs because of the way the lower bound is derived here (i.e., how the reward set and its geometry affect sample complexity). In Russo \& Vannella (2024), the complexity depends on value gaps $\Delta_r^\pi(s,a)=V_r^\pi(s)-Q_r^\pi(s,a)$ since it is a Best Policy Identification problem, whereas our analysis for Policy Evaluation involves $\rho_r^\pi(s,s')=V_r^\pi(s')-\mathbb{E}_{s''\sim P(s,\pi(s))}[V_r^\pi(s'')]$. Hence, in BPI a different analysis is needed to derive the optimization problem.
> Conceptually, our work shows how this overall strategy is more general than just identifying the best policy, and can be applied to other problems (such as policy evaluation). We will highlight this distinction more clearly in the revised version of our manuscript.
>
>
>
> >  - Could this be a potential typo?: In the line 176 second column, $\mathrm{ Alt}=\emptyset$ is empty $\to$ ${\rm Alt}$ is empty?
> > - Could this be a potential typo?: In the line 2 in Algorithm 1, $\inf \to \arg\inf$?
>
>
> Yes, both of these are indeed typos. We appreciate the reviewer for catching them and will correct them in the final version of the paper.

---

> > ### Comment · Reviewer_pv4R · 2025-04-03
> >
> > Thanks for the reply. I maintain the original positive score.

---

> > > ### Author Response · Authors · 2025-04-05
> > >
> > > Dear reviewer, thank you again for your review and for maintaining a positive assessment. We sincerely hope our clarifications have addressed your concerns. Should you have any additional questions or feedback, please do not hesitate to let us know.  With these clarifications, we hope to reinforce your overall confidence in our submission.

---

### Official Review · Reviewer_Xa3p · 2025-03-12

**Overall Recommendation:** 3

**Summary:**

This paper investigates the problem of efficiently evaluating multiple policies across multiple reward functions in an online discounted setting. The authors provide an instance-specific lower bound on sample complexity and leverages it to design an efficient exploration strategy, adapting the MR-NaS exploration scheme. The paper provides theoretical guarantees on sample complexity and demonstrates the effectiveness of the approach through experiments in tabular environments.

**Claims And Evidence:**

Yes.

**Essential References Not Discussed:**

I wonder the relationship of this work with reward-free RL. It seems that reward-free RL can potentially solve this problem, since reward-free RL can collect an informative dataset without the knowledge of reward and guarantee accurate estimation of the value functions given any reward.

**Experimental Designs Or Analyses:**

Yes, the experiments look good to me.

**Methods And Evaluation Criteria:**

Yes.

**Other Comments Or Suggestions:**

None.

**Other Strengths And Weaknesses:**

None.

**Questions For Authors:**

None.

**Relation To Broader Scientific Literature:**

The key contribution is of this paper is that this is the first work for multi-reward multi-policy evaluation. It provides a lower bound for the sample complexity and proposes a provably sample-efficient algorithm, which lays solid theoretical foundation for this topic. However, a limitation is that the discussion is only restricted in the tabular setting. With function approximation, computing $w^*_t$ can be very hard and inefficient. I would suggest the authors try finding an efficient surrogate of this step and implementing their algorithms in more complexed environments, which would make a greater impact.

**Theoretical Claims:**

Yes, I checked all the theorems, and they looked good to me.

---

> ### Author Rebuttal · Authors · 2025-03-28
>
> We sincerely thank the reviewer for their evaluation of our work and their positive remarks.
>
>
> > The key contribution is of this paper is that this is the first work for multi-reward multi-policy evaluation. [...] However, a limitation is that the discussion is only restricted in the tabular setting. With function approximation, computing $w^*_t$ can be very hard and inefficient.
>
> Indeed, we plan to extend our framework to include function approximation. However, as this is the first work specifically focused on multi-reward multi-policy evaluation, we focused on having solid theoretical results in the tabular regime  (e.g., characterizing the set of confusing models and deriving a reward-free sample-complexity bound). Furthermore, many theoretical investigations  often do not include empirical evaluations; we believe our experiments in the tabular setting offer valuable insights into the practicality of our approach and can inspire  future lines of research involving function approximation.
>
>
> > I wonder the relationship of this work with reward-free RL. It seems that reward-free RL can potentially solve this problem, since reward-free RL can collect an informative dataset without the knowledge of reward and guarantee accurate estimation of the value functions given any reward.
>
> We thank the reviewer for the opportunity to clarify this point. Note that our setting encompasses reward-free RL but not vice versa, since our problem formulation can handle both finite and convex reward sets (thus covering the entire set of rewards, as in reward-free RL). Furthermore, the instance-dependent perspective directly accounts for the complexity of the reward set, whereas reward-free RL typically does not.
>
> Moreover, while reward-free RL can indeed collect an informative dataset without knowledge of the reward, for problems with a small number of rewards, using reward-free RL may unnecessarily increase the sample complexity.
>
> Lastly, in Corollary 4.6, we provide an instance-dependent sample-complexity result for the entire set of rewards (as in reward-free RL). This result is novel, and we emphasize that such instance-dependent analyses are not common in the reward-free setting.

---

> > ### Comment · Reviewer_Xa3p · 2025-04-03
> >
> > I appreciate the authors' response and will maintain my score.

---

> > > ### Author Response · Authors · 2025-04-05
> > >
> > > Dear reviewer, thank you again for your review and feedback. We appreciate your positive assessment and hope to have addressed your concerns. If you have any further questions or require additional details, please feel free to reach out.  We hope our explanations have offered further clarity on the quality and intent of our work.

---

### Official Review · Reviewer_QAVi · 2025-03-13

**Overall Recommendation:** 3

**Summary:**

This paper addresses the problem of online multi-reward multi-policy evaluation in reinforcement learning, aiming to estimate the value functions of multiple policies across diverse reward sets with (ε, δ)-PAC guarantees. The authors derive an instance-dependent sample complexity lower bound that scales with a value deviation measure, capturing the interaction between rewards and transitions. They propose MR-NaS, an adaptive exploration algorithm that leverages this bound to guide efficient data collection. The algorithm extends prior work on Multi-Reward Best Policy Identification (BPI) and is shown to be asymptotically optimal. Experiments in tabular environments demonstrate MR-NaS's effectiveness compared to baseline methods.

**Claims And Evidence:**

See the comments below

**Essential References Not Discussed:**

See the comments below

**Experimental Designs Or Analyses:**

See the comments below

**Methods And Evaluation Criteria:**

See the comments below

**Other Comments Or Suggestions:**

NA

**Other Strengths And Weaknesses:**

Strengths:

- The work is the first to provide instance-dependent sample complexity bounds for multi-reward policy evaluation, addressing a gap in PAC guarantees for this setting.
- The analysis introduces a novel value deviation measure and connects it to sample complexity, offering insights into how reward structure impacts evaluation difficulty.
- Applications in areas like language model fine-tuning and robotics motivate the problem, highlighting real-world significance.

Weaknesses:

- The communicating MDP assumption may restrict applicability to environments with transient states.
- Experiments are limited to tabular domains; extensions to high-dimensional or continuous spaces are not discussed. Practical aspects (e.g., handling model estimation errors in non-tabular settings) are underdeveloped.

**Questions For Authors:**

- How does sample complexity scale with the number of policies/rewards? A scalability analysis would help assess broader utility.

**Relation To Broader Scientific Literature:**

See the comments below

**Theoretical Claims:**

See the comments below

---

> ### Author Rebuttal · Authors · 2025-03-28
>
> We sincerely thank the reviewer for evaluating our work and acknowledging the novelty of our paper.
>
> Below, we address your concerns in detail.
>
>
> >The communicating MDP assumption may restrict applicability to environments with transient states.
>
> We note that our focus on communicating MDPs is a standard assumptions in Best Policy Identification (see Russo \& Vannella, 2024 and Al Marjani et al., 2021). For weakly communicating MDPs, as $\delta \to 0$, transient states have negligible impact on the sample complexity asymptotically. Our main goal here is to illuminate how MDP-dependent quantities shape sample complexity for multi-reward multi-policy evaluation.
>
> > Experiments are limited to tabular domains; extensions to high-dimensional or continuous spaces are not discussed. Practical aspects (e.g., handling model estimation errors in non-tabular settings) are underdeveloped.
>
> Extending our theoretical framework and algorithms to non-tabular settings is indeed an exciting future direction. As this is the first work on multi-reward multi-policy evaluation with instance-dependent PAC guarantees, we prioritized a rigorous treatment in the tabular setting (e.g., characterizing confusing models, establishing a reward-free sample-complexity bound). Furthermore, many theoretical investigations often do not include empirical evaluations; we believe our  experiments in the tabular setting offer valuable insights into the practicality of our approach and can inspire  future lines of research involving function approximation.
>
> > How does sample complexity scale with the number of policies/rewards? A scalability analysis would help assess broader utility.
>
>
>
>
> The result in Theorem 4.5 shows that the sample complexity depends on the _worst_ reward-policy pairs in the sets of rewards and policies considered. These “worst” pairs can differ from one MDP to another—what is easy in one MDP could be difficult in another. This captures the inherent difficulty of the reward-policy sets and is not just a sum over individual pairs.
>
> For example, in the generative setting, choosing a uniform distribution $\omega(s,\pi(s)) = \frac{1}{|S|}$ yields a bound on the order of
> $
> O\left(\max_{\pi,r}  \frac{\gamma^2 |S|  \max_s ||\rho_r^\pi(s)||^2}{4 \epsilon^2 (1-\gamma)^2}\right),
> $
> and we have $\|\rho_r^\pi(s)\|_\infty = O\left(\frac{1}{1-\gamma}\right)$. Consequently, this yields a worst-case scaling of
> $
> O\left(\frac{\gamma^2|S|}{4\epsilon^2 (1-\gamma)^4}\right),
> $
> which is independent of the number of policies/rewards. We will ensure to clarify these points further in the final manuscript.

---

### Official Review · Reviewer_kVSU · 2025-03-14

**Overall Recommendation:** 3

**Summary:**

This paper studies the problem of devising an optimal data-collection policy that can evaluated policies in multi-reward and multi-policy setting. The paper adopted a PAC sample complexity perspective over finite or convex set of rewards. The analysis of the problem revolves around the set of alternative set and constructing lower bounds for related concepts. The proposed algorithm for the problem is adopted from (Russo & Vannella, 2024). Empirical results on 4 benchmark shows promising performances compared with some existing approaches.
_______________________________________
### Update after rebuttal: I appreciate the authors’ effort in rebuttal. Most of my concerns have been addressed. However, considering the manuscript would benefit from clearer exposition, in particular, with that of (Russo & Vannella, 2024), I will keep my score.

**Claims And Evidence:**

Largely yes, but some details can be provided for clarity. See comment sections for details.

**Essential References Not Discussed:**

The reviewer believes essential related works are there.

**Experimental Designs Or Analyses:**

The reviewer didn’t implement the pseudo code in the local machine to redo the experimental studies, but went over the simulation results.

**Methods And Evaluation Criteria:**

Yes.

**Other Comments Or Suggestions:**

1.	It is quite important to distinguish the work described in this work and that of (Russo & Vannella, 2024). This is because the algorithms are exactly the same, and in (Russo & Vannella, 2024), the problem is also about to identify a policy identification problem. With clear distinctions, it will position the current paper better.
2.	Around line 145, the paper mentioned the considered target policies are deterministic policies. Will analysis for stochastic policy be significantly different?
3.	In Line 211, it says confusing set will be close to M (in the KL sense). However, in the definition of the set of alternative models, only shows distant way by 2$\epsilon$ in infinite norm sense. Please clarify these concepts.
4.	In Line 219, it’s unclear on what does it mean that $P(s,a)$ is continuous w.r.t. $P’(s,a)$ as $P(s,a)$ is not a function of $P’(s,a)$. Please elaborate.
5.	The definition for $P’_r(s,a)$ is unclear.
6.	Typo: Line 22 right column, different task -> different tasks
7.	Typo: Line 69, a period is left out.

**Other Strengths And Weaknesses:**

Strength: The paper poses an interesting question of finding an efficient sampling strategy in the setting of multi-reward and multi-policy. The empirical results show promising results over existing algorithms.

Weakness: See the comment section below.

**Questions For Authors:**

Please see the comment section.

**Relation To Broader Scientific Literature:**

The paper studied policy evaluation reinforcement learning in the setting of multi-reward and multi-policy setting. It poses an interesting question of finding an efficient sampling strategy under such setting. The problem seems to be general and contains traditional RL as a special case.

**Theoretical Claims:**

No.

---

> ### Author Rebuttal · Authors · 2025-03-28
>
> We appreciate the reviewer’s  feedback and the time they dedicated to evaluating our paper. Below, we provide detailed responses to each of their concerns.
>
> > It is quite important to distinguish the work described in this work and that of (Russo \& Vannella, 2024). [...] With clear distinctions, it will position the current paper better.
>
> We thank the reviewer for the comment. Prior work has focused on the Best Policy Identification (BPI) problem, but the technique is far more general and can be applied to other problems. Our contribution is to show how the same broad approach can be applied to the multi-reward, multi-policy evaluation setting.
>
> The overall technique is inspired by approaches in Best Arm Identification (Garivier \& Kaufmann, 2016), where the problem is cast as a hypothesis-testing framework to distinguish the true model from the "worst'' confusing model (see also Degenne \& Koolen, 2019; Al Marjani et al., 2021). This perspective permits to compute an exploration strategy $w_t^*$ at each time step that maximizes the evidence gathered from the model (depending on the set of possible alternative models).
>
> While the general algorithmic structure is similar in all these works, the specific optimization problem in computing $w_t^*$ differs because of the way the lower bound is derived here (i.e., how the reward set and its geometry affect sample complexity). In BPI, the complexity depends on value gaps $\Delta_r^\pi(s,a)=V_r^\pi(s)-Q_r^\pi(s,a)$, whereas our analysis involves $\rho_r^\pi(s,s')=V_r^\pi(s')-\mathbb{E}_{s''\sim P(s,\pi(s))}[V_r^\pi(s'')]$. Hence, in BPI a different analysis is needed to derive the optimization problem.
> We will clarify these distinctions more explicitly in the final version of the paper.
>
>
> > Around line 145, the paper mentioned the considered target policies are deterministic policies. Will analysis for stochastic policy be significantly different?
>
> The analysis can be carried out with stochastic policies in a similar way (in the appendix we also have results for stochastic policies), and we will point this out in the paper. We chose to mainly work with deterministic policies to avoid overly cumbersome notation.
>
> > In Line 211, it says confusing set will be close to $M$ (in the KL sense). However, in the definition of the set of alternative models, only shows distant way by $2\epsilon$ in infinite norm sense. Please clarify these concepts.
>
> We thank the reviewer for the question and the opportunity to clarify this point. The set of alternative models $Alt_{\pi,r}^\epsilon$ contains all models $M'$ satisfying $\|V_{M_r}^\pi - V_{M_r'}^\pi\|_\infty > 2\epsilon$. The sample complexity is characterized by the ``worst'' model in this set, i.e., the one that maximizes the sample complexity. This is exactly what we do in Eq. (2): we solve an optimization problem that tries to find an alternative model that minimizes the KL-divergence between the transition function $P$ of $M$ and the transition function $P_r'$ of an alternative model $M_r'$. We will make sure to clarify this point in the paper.
>
> > In Line 219, it’s unclear on what does it mean that $P(s,a)$ is continuous w.r.t. $P'(s,a)$ as $P(s,a)$ is not a function of  $P'(s,a)$. Please elaborate.
>
> What we mean is that $P(s,a)$ is an absolutely continuous measure with respect to $P'(s,a)$. In other words, for all $s'$ such that $P'(s'|s,a) = 0$, we have $P(s'|s,a)=0$. We will clarify this in the paper.
>
> > The definition for $P_r'(s,a)$ is unclear.
>
> We understand that this definition may seem odd. For an alternative model $M_r'$, we denote by $P_r'$ its transition function(the subscript $r$ is only added to highlight for which reward $r$ the transition function induces an alternative model). We will explicitly explain this notation in the paper.
> > Typo: Line 22 right column, different task $\to$ different tasks.  Typo: Line 69, a period is left out.
>
> We appreciate the reviewer pointing out these typos. We will correct them in the paper.

---

> > ### Comment · Reviewer_kVSU · 2025-04-08
> >
> > I would like to thank the authors for the clarification and responses. I would like to keep my score.

---

> > > ### Author Response · Authors · 2025-04-08
> > >
> > > We sincerely thank the reviewer again for their feedback and hope our clarifications have addressed your concerns. If there remain any specific suggestions or additional improvements that could help strengthen the manuscript further, and potentially lead you to reconsider your score upward, we would be very grateful to know. We remain committed to further improving our manuscript.

---

### Decision · Program_Chairs · 2025-05-01

**Decision:**

Accept (poster)

**Comment:**

This paper studies a policy evaluation problem where we wish to evaluate multiple policies under multiple rewards simultaneously, and give a sample complexity bound for simultaneous evaluation which depends on the policy and reward sets, capturing improvements over the naive baseline of independent evaluation. All the reviewers are positive about the work, and I agree with their assessment. I encourage the authors to surface the key reviewer questions and the ensuing discussions in the final revision of the paper.